# STRUCTURED-NOISE MASKED MODELING FOR VIDEO, AUDIO AND BEYOND

## ABSTRACT

Masked modeling has emerged as a robust self-supervised learning framework. However, most methods rely on random masking, which disregards the structural properties of different data modalities. To naturally align with the spatiotemporal and spectral characteristics of video and audio data, we introduce a structured noise-based masking approach. By filtering white noise into different color noise distributions, we generate structured masks that capture modality-specific patterns without requiring handcrafted heuristics or access to the data. Our approach enhances masked video and audio modeling frameworks without any additional computational cost. Experiments show that structured noise masking consistently outperforms random masking, underscoring the value of modality-aware masking strategies for representation learning.

## 1 INTRODUCTION

Self-supervised masked modeling has emerged as a powerful paradigm for learning representations across modalities, such as images (He et al., 2022; Bao et al., 2021; Li et al., 2023), videos (Tong et al., 2022; Salehi et al., 2024; Thoker et al., 2025), and audio (Baevski et al., 2022; Huang et al., 2022; Chong et al., 2023). The core idea is to mask portions of the input, whether patches, spectrogram regions, or spatiotemporal tubes, and train a model to reconstruct them. This mask-and-predict objective compels the network to capture rich contextual information without labels. The standard approach is to use random masking, which uniformly drops tokens and achieves strong performance for its simplicity and generality. However, random masking disregards modality-specific inductive biases: images are spatially coherent, videos exhibit spatiotemporal continuity, and audio spectrograms follow frequency–time patterns.

To overcome random masking limitations, alternative masking strategies (Madan et al., 2024; Bandara et al., 2023; Hernandez et al., 2024; Huang et al., 2023b; Kara et al., 2024) have been explored that better align with the intrinsic structures of data modalities. For example, in image modeling, recent efforts have explored adaptive masking, where masks are guided by semantics, attention, or motion cues (Li et al., 2022; Chen et al., 2023; Bandara et al., 2023; Fan et al., 2023; Huang et al., 2023a). While effective, these approaches often depend on predefined heuristics or auxiliary networks (Li et al., 2022), or costly computations (e.g., motion priors (Fan et al., 2023; Huang et al., 2023a), adversarial training (Chen et al., 2023)), which reduces their flexibility and scalability across modalities.

An alternative direction, and the one we pursue in this paper, is to generate structured masks from filtered noise distributions. This idea was first introduced for static images in ColorMAE (Hinojosa et al., 2024), where white noise was filtered into distinct 2D color noise types to produce data-independent masks. However, its scope is limited to static 2D inputs, leaving open the question of whether color noise can be extended to video masking, where masks must evolve smoothly over time, and to audio, where spectrogram masks should respect frequency–time coverage. In this work, we extend structured noise masking beyond static images, introducing video and audio variants designed for each modality's structure. We propose modality-specific noise filters that generate masks aligned with each modality's inductive biases, enhancing self-supervised learning via the mask-and-predict objective. We summarize our contributions as:

- **3D Green masking (Green3D) for video:** a color noise–based strategy that produces spatiotemporally coherent masks, preserving spatial clustering and temporal smoothness.

- **Regularized Blue noise (R-BN) masking for audio:** a 2D blue-noise-based optimization algorithm that produces masks by enforcing a controlled distribution of visible spectrogram patches across time and frequency, leveraging the structure of audio.
- **Multimodal evaluation:** Through extensive experiments across diverse benchmarks, including Kinetics-400 and SomethingSomething V2 for video, AudioSet-20K and ESC-50 for audio, and VGG-Sound for joint audio-video, we show that our proposed masking consistently boosts the performance of existing self-supervised methods in video-only, audio-only, and multimodal settings.

To our knowledge, our method is the first data-independent masking strategy to consistently outperform random masking across video, audio, and multimodal settings without added computation overhead.

## 2 RELATED WORK

**Masked Modeling for Video.** Masked modeling is a widely used self-supervised paradigm that originated with masked language modeling in BERT (Devlin et al., 2019), and was later extended to audios with wav2vec (Schneider et al., 2019; Baevski et al., 2020) and vision by MAE (He et al., 2022; Xie et al., 2022b). The main idea is to randomly mask a portion of the input data and then reconstruct it with an asymmetric encoder-decoder network. VideoMAE (Tong et al., 2022) extends MAE to videos using spatiotemporal random tube masking. Most follow-up works still rely on random masking but with slightly different architectures (Girdhar et al., 2023; Sun et al., 2023; Lu et al., 2023; Salehi et al., 2024; Thoker et al., 2025). For example, SIGMA (Salehi et al., 2024) refines the learning objective by guiding reconstruction with a Sinkhorn-based matching loss. SMILE (Thoker et al., 2025) infuses both spatial and motion semantics by leveraging image-language pretrained models such as CLIP. Although effective, these methods often increase computational overhead and parameter count due to architectural complexity.

**Masked Modeling for Audio.** Masked modeling has also been explored in audio. wav2vec 2.0 (Baevski et al., 2020) masks latent speech units with a contrastive loss, while Data2Vec (Baevski et al., 2022) unifies masked prediction across modalities. SS-AST (Gong et al., 2022a) adapts masked spectrogram prediction to semi-supervised audio tasks, MaskSpec (Chong et al., 2023) proposes spectrogram-aware masking, and MAE-AST (Baade et al., 2022) applies MAE to audio spectrogram transformers. Audio-MAE (Huang et al., 2022) adapts MAE to spectrograms at scale and achieves competitive performance. In addition, multimodal extensions such as MST (Li et al., 2021) explore synchronized tube masking across audio and video streams, while MultiMAE (Bachmann et al., 2022) leverages shared latent tokens and cross-attention to jointly reconstruct multiple modalities. However, these approaches typically rely on learned or cross-modal masking mechanisms rather than predefined, data-independent patterns. Despite these advances, most works still rely on random masking, which may discard semantically important or structurally coherent spectrogram regions.

**Data-driven Masking.** These methods adapt the masking process itself, using data semantics, attention, or motion cues. For images, attention-guided masking (Sick et al., 2025) selects informative patches, SemMAE (Li et al., 2022) masks semantic parts, and CL-MAE (Madan et al., 2024) applies curriculum masking. For video, AdaMAE (Bandara et al., 2023) introduces an adaptive masking strategy that selects informative spatiotemporal patches, enabling effective pretraining even at high masking ratios. Also, MGMAE (Huang et al., 2023a) and MGM (Fan et al., 2023) guide the masking with optical flow or motion vectors. In audio, spectrogram-aware masking (Chong et al., 2023) adapts to time-frequency context, while MAViL (Huang et al., 2023b) balances masking across modalities. Although effective, these methods rely on external supervision, such as flow estimation, pretrained semantics, or reinforcement learning, which increases complexity. This motivates alternatives that achieve competitive performance without incurring additional computational cost.

**Data-independent Masking.** Unlike adaptive methods, data-independent approaches replace random masking with deterministic rules that capture spatial, temporal, or spectral structure without extra supervision. For images, BEiT (Bao et al., 2021) explored block- or grid-based masking to enforce uniform coverage. In speech, SpecAugment (Park et al., 2019) masks time or frequency bands. Data-independent frequency-based masking variants include MFM (Xie et al., 2022a), FMAE (Liu

et al., 2024) for multimodal biosignals, and iBOT (Zhou et al., 2021) focusing on high-frequency content. More recently, ColorMAE (Hinojosa et al., 2024) introduced color noise–based masking for images, filtering white noise with low-pass, band-pass, or high-pass filters to generate structured masks that are data-independent, efficient, and better aligned with localized spatial patterns. While effective, it remains restricted to static 2D inputs and does not address temporal coherence in video, spectral structure in audio, or complementary masking needs in multimodal learning. Building on color masking, we propose **Green3D masking** for spatiotemporal video and **Regularized Blue Noise (R-BN) masking** for spectrogram-aware audio, further combining them for multimodal setups. This yields simple, training-free masking strategies designed for different modalities, avoiding motion priors or auxiliary supervision while remaining competitive with adaptive methods.

## 3 METHODOLOGY

### 3.1 PRELIMINARIES

**Masked Modeling.** In general, an input $X$ (image, video, or audio spectrogram) is partitioned into patches and embedded into a sequence of token representations via a function $\phi$, yielding $X_p = \phi(X) \in \mathbb{R}^{N \times d}$, where $N$ is the number of patches and $d$ the embedding dimension. A binary mask $M \in \{0, 1\}^N$ is generated by a masking function $\eta$ with a mask ratio $\gamma \in [0, 1]$, using random noise $n_w \sim \mathcal{N}(0, 1)$: $M = \eta(X_p, n_w, \gamma)$. Each entry $M_i$ corresponds to patch $i$, with $M_i = 1$ if the patch is visible and $M_i = 0$ if masked. The token sets are

$$X_p^{\text{visible}} = X_p \odot M \quad , \quad X_p^{\text{masked}} = X_p \odot (1 - M),$$

where $\odot$ denotes the Hadamard product. The encoder processes $X_p^{\text{visible}}$, while the decoder reconstructs the full input as $\hat{X}$ by integrating both visible and masked tokens. Training minimizes mean squared error: $\mathcal{L}_{\text{recon}} = |X - \hat{X}|_2^2$. Random masking is widely used due to its simplicity (He et al., 2022; Tong et al., 2022; Huang et al., 2022), but it ignores modality-specific inductive biases such as spatial coherence in images, temporal continuity in videos, and spectral patterns in audio.

**Color Noise Masking.** Instead of random masking, *structured noise* can be leveraged to introduce modality-specific masks. Unlike white noise, with a uniform power distribution across all frequencies, filtering it through frequency constraints produces *structured noise patterns* that align with spatial and temporal structures (Lau et al., 2003; Correa et al., 2016). Given white noise $n_w$, frequency-constrained noise is constructed by filtering it with a $d$-dimensional Gaussian kernel $G_\sigma(\mathbf{x}) = (2\pi)^{-d/2} \sigma^{-d} \exp(-\|\mathbf{x}\|^2/(2\sigma^2))$, where $\mathbf{x} \in \mathbb{R}^d$ are spatial coordinates and $\sigma$ controls the filter scale. Convolving $n_w$ with $G_\sigma$ of different bandwidths yields distinct noise components:

$$n_r = G_\sigma * n_w, \quad (1) \qquad n_b = n_w - G_\sigma * n_w, \quad (2) \qquad n_g = G_{\sigma_1} * n_w - G_{\sigma_2} * n_w, \quad (3)$$

where $*$ denotes convolution and $\sigma_1 < \sigma_2$. These noise patterns define the structured masks: $M_r = \eta(X_p, n_r, \gamma)$, $M_b = \eta(X_p, n_b, \gamma)$, and $M_g = \eta(X_p, n_g, \gamma)$. The precise definition of $\eta$ is provided in Algorithm 2 in the Appendix A.11.

As shown in Fig. 1 for $d = 2$, red noise ($n_r$) preserves low frequencies, producing smooth, large-scale masks; blue noise ($n_b$) enhances high-frequency details, creating fine-grained masks; green noise ($n_g$) balances both, generating mid-sized, clustered masks. These structured masks encourage the model to learn features aligned with spatial frequency. In the following sections, we extend this principle beyond images to *video (Green3D)* and *audio (Regularized Blue)* data.

### 3.2 GREEN 3D NOISE FOR VIDEO MASKING

Video masking should respect both spatial coherence and temporal continuity. Standard methods, such as VideoMAE (Tong et al., 2022) and SIGMA (Salehi et al., 2024), rely on random tube masking, which applies a static mask across frames, preserving temporal consistency but lacking adaptability to motion. To address this, we propose *Green3D Noise Masking*, which introduces structured, evolving masks across frames, enhancing fine-grained temporal representation learning.

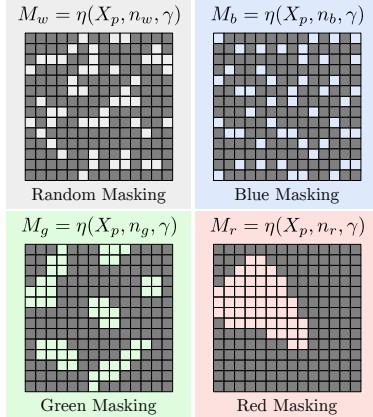

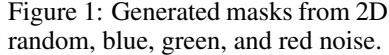

Figure 1: Generated masks from 2D random, blue, green, and red noise.

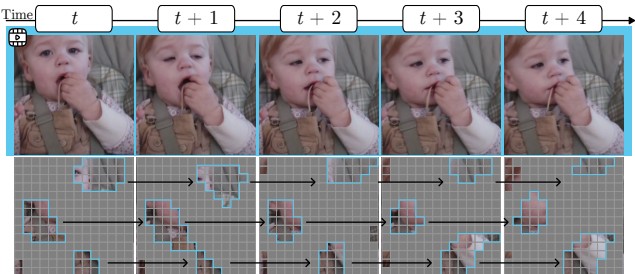

Figure 2: Unlike traditional random tube masking, which enforces strict temporal consistency, our proposed Green 3D masking generates structured random masks that evolve smoothly across consecutive frames. This smooth evolution prevents abrupt masking changes, enabling the model to better capture natural temporal dynamics and continuity in video data.

**Green3D Mask Generation.** We generate the proposed noise for video masking by applying equation 3 with a 3D white noise tensor $n_w$ and two 3D Gaussian kernels $G_\sigma$ ($d{=}3$) defined as:

$$G_\sigma(\mathbf{x}) = \frac{1}{(2\pi)^{\frac{3}{2}}\sigma^3} \exp\left(-\frac{\|\mathbf{x}\|^2}{2\sigma^2}\right),\qquad(4)$$

where $\mathbf{x}{=}(x,y,z) \in \mathbb{R}^3, \sigma \in \{\sigma_1, \sigma_2\}$, and $\sigma_1{<}\sigma_2$ control the frequency response. A smaller $\sigma_1$ preserves fine details, while a larger $\sigma_2$ removes high-frequency components. Instead of fixing $(\sigma_1, \sigma_2)$, which produces masks with limited diversity and either overly static or noisy temporal behavior, we randomly sample $(\sigma_1, \sigma_2)$ within a bounded range $[0.5, 2]$ for each sequence. This generates an ensemble of 3D green noise tensors, encouraging diverse but still mid-frequency spatiotemporal patterns. Empirically, this stochasticity yields smoother mask evolution and prevents overfitting to a single frequency, making the approach more robust than a fixed parameterization. Our Green3D masks are obtained as $\eta(X_p, n_g^{3D}, \gamma)$, where $\eta$ follows the masking function in (He et al., 2022; Hinojosa et al., 2024). As seen in Fig. 2, our Green3D masks evolve smoothly over time, avoiding abrupt frame-to-frame changes and enabling the model to better learn temporal continuity.

### 3.3 REGULARIZED BLUE NOISE FOR AUDIO MASKING

Self-supervised audio learning relies on spectrogram representations, where structured time–frequency patterns encode meaningful information. However, random masking (Huang et al., 2022) misaligns with the inherent structure of spectrograms. As seen in Fig. 1, random, green, and red noise masking create clusters of visible patches or large masked regions. While beneficial in vision tasks, these clusters do not correspond to meaningful time-frequency events in audio. Instead, blue noise masking offers a more effective strategy, as it distributes visible patches more evenly.

Blue noise patterns have been widely studied in computer graphics and image processing (Wolfe et al., 2022; Rauhut, 2010; Ahmed & Wonka, 2020; Correa et al., 2016) for their ability to suppress low-frequency components. A simple way to generate blue noise is by filtering white noise via a Gaussian kernel, as in equation 2. However, this does not explicitly control the separation between visible patches, still leading to small clusters. To overcome this, we introduce an optimization-based approach that enforces spatial separation constraints to ensure a better distribution of visible patches. This leads to our proposed *Regularized Blue Noise (R-BN)* Masking, ensuring a more well-distributed masking pattern for spectrogram-based audio representations.

**Regularized Blue Noise (R-BN) Mask Generation.** We generate an initial set of $K$ candidate masks $\{M^i\}_{i=1}^K$ by thresholding noise values from $n_w$ or $n_b$ at the target ratio $\gamma$. Each mask is then iteratively optimized to maintain uniform patch separation. For each spatial position $P{=}(x,y)$, processed in randomized order, we consider a local window $U_P^i \in \mathbb{R}^{\Delta \times \Delta}$ centered at $P$ for each candidate $M^i$. To measure clustering, we count visible patches aligned with $P$ along four orientations—horizontal ($d_1^i$), vertical ($d_2^i$), and the two diagonals ($d_3^i, d_4^i$). The clustering score is:

$$S_P^i = w_1 d_1^i + w_2 d_2^i + w_3 d_3^i + w_4 d_4^i,\qquad(5)$$

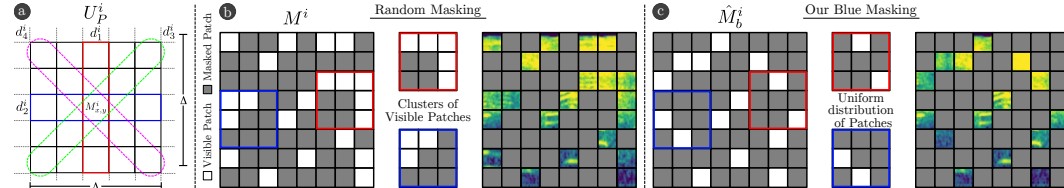

Figure 3: **ⓐ** Illustration of the metric used to determine the concentration of visible patches in a window $U_P^i$ of the mask $M_{x,y}^i$. **ⓑ** Example of the initial mask ($M^i$), with clusters of visible patches, and **ⓒ** final mask ($\hat{M}_b^i$) obtained with our Regularized blue noise masking algorithm, with uniformly distributed visible patches. Note the improved uniformity in the final mask, which ensures better coverage and reduces undesirable clustering effects.

where $w_1, w_2, w_3, w_4$ weight the directional counts (set to 1 by default). The candidate with the lowest score is selected as $\hat{i} = \arg\min_i S_P^i$. The patch update is then applied:

$$\hat{M}_{x,y}^i = \begin{cases} 1, & \text{if } i = \hat{i} \text{ (visible)}, \\ 0, & \text{otherwise (masked)}. \end{cases} \quad (6)$$

This process continues until the number of visible patches in each mask reaches $(1 - \gamma)N$, ensuring the target masking ratio. The optimized masks $\hat{M}_b$ are retained for training, providing more uniform and well-separated visible patches than standard blue noise masks $M_b$. As shown in Fig. 3, this procedure reduces local clustering effects observed in prior work (Hinojosa et al., 2024). Empirically, we find that R-BN improves representation learning on audio tasks, particularly in settings where time–frequency coverage is crucial. Full pseudocode is provided in Appendix A.10.

Our design choices for the masks are motivated by modality-specific structural properties: videos exhibit mid-frequency spatiotemporal statistics, whereas audio spectrograms contain fine-grained time–frequency harmonic patterns with energy concentrated in low-to-mid frequencies. We provide a frequency-domain analysis in Appendix A.15 that supports these structured mask choices.

### 3.4 Blue & Green Noise for Audio-Visual Masking

Our color masking extends naturally to joint audio-video setups. Specifically, we apply Green masking $M_g$ to video frames for structured spatial masking with frame-wise consistency, while Distributed Blue noise masks $\hat{M}_b$ enforce a uniform distribution of visible patches in audio. This modality-specific masking better aligns with the structural properties of each domain, enhancing joint pretraining within a unified masked modeling framework.

Notably, all our proposed masks, including Green3D and R-BN, are precomputed as mask tensors (e.g., Green3D masks of shape $N \times 64 \times 64 \times 64$). During training, they undergo standard augmentations such as random flipping, normalization, and resizing to match the input volume (e.g., $14 \times 14 \times 8$), preserving their noise properties while providing diverse and efficient structured masking without any added computational overhead.

## 4 Experiments

### 4.1 Results for Video

**Evaluated Architectures.** We evaluate two masked video modeling frameworks: VideoMAE (Tong et al., 2022), which reconstructs raw video pixels from masked inputs, and SIGMA (Salehi et al., 2024), which instead predicts DINO features of masked frames with a Sinkhorn-cluster matching loss. In both cases, we replace the default random tube masking with our proposed Green3D masking, keeping the remaining architecture unchanged.

**Implementation Details.** Following prior setups (Tong et al., 2022; Salehi et al., 2024), we adopt an encoder–decoder framework with a ViT-B backbone and identical pretraining hyperparameters. Unless otherwise specified, models are pretrained on **Kinetics-400** (Kay et al., 2017) and **Something-Something V2** (Goyal et al., 2017), consistent with the standard masked video modeling protocol (Tong et al., 2022; Sun et al., 2023; Fan et al., 2023; Huang et al., 2023a). After pretraining,

Table 1: **Comparison of masked video modeling methods on Something-Something V2 and Kinetics-400 for standard action recognition.** All results use a ViT-B backbone pretrained on K400 or SSv2 for 800 epochs. Our Green3D masking consistently improves both in-domain and cross-domain performance for VideoMAE as well as advanced architectures such as SIGMA.

| | Masking Type | | SSv2 Pretraining | K400 Pretraining | |
|---|---|---|---|---|---|
| Method | Data-independant | Data-adaptive | SSv2 Top-1 | SSv2 Top-1 | K400 Top-1 |
| VideoMAE | Random | - | 69.6 | 68.5 | 80.0 |
| VideoMAE + Our masking | Green3D | - | 70.8$^{(+1.2\%)}$ | 69.7$^{(+1.2\%)}$ | 80.5$^{(+0.5\%)}$ |
| CMAE-V | Random | - | 69.7 | - | 80.2 |
| OmniMAE | Random | - | 69.5 | 69.0 | 80.8 |
| MME | Random | - | 70.0 | 70.5 | 81.5 |
| MGM | - | Motion | 70.6 | 71.1 | 80.8 |
| MGMAE | - | Motion | 71.0 | 68.9 | 81.2 |
| SIGMA | Random | - | 71.2 | 71.1 | 81.5 |
| SIGMA + Our masking | Green3D | - | 72.0$^{(+0.8\%)}$ | 71.8$^{(+0.7\%)}$ | 82.1$^{(+0.6\%)}$ |

the decoder is discarded and the encoder backbone is finetuned for downstream tasks. Additional details are provided in the Appendix A.12.

### 4.1.1 STANDARD ACTION RECOGNITION

**Datasets.** Following prior works (Tong et al., 2022; Sun et al., 2023; Fan et al., 2023; Huang et al., 2023a), we evaluate on two standard action recognition benchmarks: **Kinetics-400 (K400)** (Kay et al., 2017) and **Something-Something V2 (SSV2)** (Goyal et al., 2017). K400 comprises 240k training and 20k validation videos across 400 human action categories, emphasizing spatial and object-centric cues. In contrast, SSv2 contains 169k training and 25k validation clips spanning 174 categories, focusing on fine-grained temporal interactions, which makes it a more challenging benchmark for video self-supervised learning. We follow the finetuning and evaluation protocol of (Tong et al., 2022) and report top-1 accuracy on both datasets.

**Something-Something Results.** We evaluate SSv2 under two settings: (i) *in-domain*, where models are pretrained and finetuned on SSv2, and (ii) *cross-domain*, where pretraining is on K400 and finetuning on SSv2. Results are reported in Table 1, alongside state-of-the-art masked video modeling methods.

Our Green3D masking improves VideoMAE by +1.2% in both in-domain (SSv2→SSv2) and cross-domain (K400→SSv2) transfer, validating its effectiveness over random tube masking for learning richer spatiotemporal representations. We attribute these gains to the harder mask-reconstruction patterns generated by Green3D, which encourage stronger spatiotemporal modeling.

Notably, VideoMAE+Green3D matches or surpasses motion-guided approaches such as MG-MAE (Huang et al., 2023a) and MGM (Fan et al., 2023). Specifically, it outperforms MGM in the in-domain setting and MGMAE in the cross-domain setting. Unlike these data-adaptive methods, which require access to motion priors (e.g., optical flow or motion vectors) and add substantial computational overhead (MGMAE is ∼1.5× slower than VideoMAE), Green3D is data-independent and incurs no extra cost, as masks are precomputed.

Finally, Green3D also benefits recent frameworks like SIGMA (Salehi et al., 2024), yielding +0.8% in-domain and +0.7% cross-domain, highlighting its versatility as a lightweight, plug-and-play masking strategy for current and future video models.

**Kinetics results.** For K400, we evaluate the in-domain pretraining setting in Table 1 following prior works. Similar to SSv2, our Green3D masking yields consistent gains over random tube masking (+0.5% for VideoMAE and +0.6% for SIGMA), showing that our method also enhances spatial semantics, which is crucial for datasets like K400, where many actions are distinguished by spatial cues.

Table 2: **Comparison of masked video methods for unsupervised video object segmentation.** Following, evaluation protocol from Salehi et al. (2023) we report mIoU for clustering and over-clustering. We evaluate the ViT-B backbone pretrained on K400 and use the officially released checkpoints for all prior works. When we equip VideoMAE with our masking, we significantly improve its performance and even surpass motion-guided masking methods. Our masking also boosts the performance of SIGMA when added as a plugin.

| | Clustering | | Overclustering | |
|---|---|---|---|---|
| **Method** | **YTVOS** | **DAVIS** | **YTVOS** | **DAVIS** |
| VideoMAE | 34.1 | 29.5 | 61.3 | 56.2 |
| VideoMAE + Our masking | 35.6$^{(+1.5\%)}$ | **38.2**$^{(+8.7\%)}$ | 62.5$^{(+1.5\%)}$ | 58.2$^{(+2.0\%)}$ |
| MGM | 36.6 | 36.5 | 61.2 | 56.6 |
| MGMAE | 34.5 | 31.0 | 60.1 | 57.5 |
| SIGMA | 41.1 | 33.1 | 67.1 | 59.0 |
| SIGMA + Our masking | **42.1**$^{(+1.3\%)}$ | 34.2$^{(+1.2\%)}$ | **68.4**$^{(+1.3\%)}$ | **60.0**$^{(+1.0\%)}$ |

Table 3: **Comparison on the SEVERE benchmark** (Thoker et al., 2022) evaluating domain shift, sample efficiency, action granularity, and task shift of learned video representations. All methods are pretrained on K400 with a ViT-B backbone. Our method improves the downstream generalization of both VideoMAE and SIGMA architectures.

| **Method** | **Domain shift** | | **Sample efficiency** | | **Action granularity** | | **Task shift** | | **Mean** |
|---|---|---|---|---|---|---|---|---|---|
| | SSv2 | Gym99 | UCF($10^3$) | GYM($10^3$) | FX-S1 | UB-S1 | UCF-RC↓ | Charades | |
| VideoMAE | 68.6 | 86.6 | 74.6 | 25.9 | 36.6 | 74.3 | 0.172 | 17.2 | 58.3 |
| VideoMAE + Our masking | 69.7 | 88.1 | 75.0 | 29.9 | 38.6 | 74.8 | 0.170 | 18.2 | 59.6 |
| MVD | 70.0 | 82.5 | 67.1 | 17.5 | 31.3 | 50.5 | 0.184 | 16.1 | 52.1 |
| MGMAE | 68.9 | 87.2 | 77.2 | 24.1 | 33.7 | 79.5 | 0.181 | 17.9 | 58.8 |
| MGM | 71.1 | 89.1 | 78.4 | 26.4 | 38.6 | 86.9 | **0.152** | 22.5 | 62.2 |
| MME | 70.1 | 89.7 | 79.2 | 29.8 | 55.5 | **87.2** | 0.155 | 23.6 | 65.0 |
| SIGMA | 70.9 | 89.7 | 84.1 | 28.0 | 55.1 | 79.9 | 0.169 | 23.1 | 64.2 |
| SIGMA + Our masking | **71.8** | **90.7** | **85.0** | **35.0** | **56.5** | 85.9 | 0.163 | **27.5** | **67.3** |

### 4.1.2 UNSUPERVISED VIDEO OBJECT SEGMENTATION

**Setup.** Following Salehi et al. (2024), we evaluate the spatial, temporal, and semantic representations learned by our method on the unsupervised video object segmentation benchmark from Salehi et al. (2023). Unlike action recognition, which pools space–time features into a global clip representation, this benchmark assesses the encoder's ability to produce temporally consistent segmentation maps. Space–time features are clustered using $k$-means with a predefined cluster count $K$ and aligned with ground-truth masks via the Hungarian algorithm (Kuhn, 1955). Segmentation quality is measured by mean Intersection over Union (mIoU), with two evaluation modes: *clustering* when $K$ matches the ground-truth object count, and *overclustering* when $K$ exceeds it. We report results on **DAVIS** (Pont-Tuset et al., 2017) and **YTVOS** (Xu et al., 2018). Dataset and evaluation details are provided in the Appendix A.12.

**Results.** As shown in Table 2, equipping VideoMAE with Green3D masking yields substantial improvements across all settings. On DAVIS clustering, it improves VideoMAE by +8.7% and surpasses MGMAE by +7.2%, demonstrating stronger object awareness and spatiotemporal representations. Consistent gains on YTVOS further validate its effectiveness. Since our setup matches MGMAE and MGM except for the masking strategy, these results confirm that Green3D better preserves spatiotemporal object continuity. Again, applying Green3D to SIGMA improves performance, highlighting its ability to generalize across pretraining frameworks and downstream tasks.

### 4.1.3 SEVERE GENERALIZATION BENCHMARK

**Setup.** Following recent video SSL works (Thoker et al., 2022) (Thoker et al., 2023) (Salehi et al., 2024), we further assess the robustness and generalization of our learned video representations on the SEVERE benchmark (Thoker et al., 2022), a comprehensive suite comprising eight experiments across four axes of generalization: *domain shift*, *sample efficiency*, *action granularity*, and *task shift*. Domain shift is assessed on **SomethingSomething v2** and **FineGym (Gym99)** (Shao et al., 2020), which diverge from the Kinetics-400 pretraining domain. Sample efficiency is evaluated through low-shot action recognition with only 1,000 finetuning samples, denoted as **UCF**$(10^3)$ and **Gym**$(10^3)$. Action granularity examines fine distinctions between semantically related categories via **FX-S1** and **UB-S1** splits of FineGym, where differences hinge on subtle gymnastic elements such as "jump" or "balance" variations. Finally, task shift evaluates out-of-distribution tasks, namely temporal repetition counting on **UCFRep** Zhang et al. (2020) and multi-label recognition on **Charades** Sigurdsson et al. (2016). We adhere to the original SEVERE training and evaluation protocols, with further details provided in the Appendix A.12.

**Results.** Table 3 summarizes the comparison with recent self-supervised methods on the SEVERE benchmark. For *Domain Shift*, we observe higher accuracy on SSv2 and Gym99 relative to the respective baselines, suggesting stronger robustness to unseen domains. Under *Sample Efficiency*, VideoMAE+Ours outperforms motion-guided approaches MGM and MGMAE on **Gym**$(10^3)$, while SIGMA+Ours achieves a 7% gain over the SIGMA baseline, indicating effective adaptation in low-shot settings. In the *Action Granularity* evaluation, our method yields better results on FX-S1 and UB-S1 for both frameworks, reflecting improved handling of fine-grained motion categories. For *Task Shift*, our approach maintains competitive performance on repetition counting and improves the performance on Charades multi-label recognition. Note that for UCF-RC, lower values indicate better repetition-counting accuracy, as the metric reports mean counting error. *Overall.* On average, green3D masking increases the SEVERE mean score by 1.3% with VideoMAE and by 3% with SIGMA. These improvements across all SEVERE factors demonstrate that our modality-aware masks generalize well beyond the training distribution, strengthening cross-domain and long-horizon robustness.

## 4.2 RESULTS FOR AUDIO

**Setup.** We evaluate our proposed **Regularized Blue Noise (R-BN)** masking within both AudioMAE (Huang et al., 2022) and MaskSpec (Chong et al., 2023), replacing their default random masking while leaving the rest of the framework unchanged. Following the AudioMAE protocol, we use a ViT-B backbone, apply an 80% masking ratio during pretraining and 30% during finetuning, and discard the decoder after pretraining. Pretraining is conducted on the large-scale **AudioSet-2M** (Gemmeke et al., 2017), which contains over 2M audio clips spanning a diverse range of sound events, including non-speech audio. For downstream evaluation, the pretrained encoder is finetuned separately on **AudioSet-20K (AS-20k)**, a balanced 20k subset of AudioSet for efficient benchmarking; **ESC-50** (Piczak, 2015), consisting of 2,000 environmental sound recordings across 50 classes; and **SPC-2** (Warden, 2018), which provides additional coverage of sound classification tasks. Note that the publicly available AudioSet test split has changed over time due to YouTube removals, hence the ∗; we therefore evaluate both baselines and our method on the same current split to ensure a fair comparison. More training and evaluation details are provided in the Appendix A.13. This setup enables us to assess the effect of R-BN across both large-scale pretraining and fine-grained audio recognition tasks, ensuring fair comparison with prior baselines.

**Results.** Table 4 compares AudioMAE (Huang et al., 2022) and MaskSpec (Chong et al., 2023) with their R-BN variants, alongside other self-supervised audio baselines. Our masking method consistently improves performance across all benchmarks: for AudioMAE, R-BN yields +0.7% on AS-20k, +0.9% on AS-2M, +0.5% on ESC-50, and +0.4% on SPC-2; for MaskSpec, the gains are even larger, reaching +1.1%, +0.5%, +0.8%, and +0.5%, respectively. These results confirm that R-BN enhances both frameworks, outperforming prior baselines without requiring additional supervision or heuristics. Unlike MaskSpec, which relies on predefined time–frequency rules, or MAE-AST (Baade et al., 2022), which benefits from extra speech data, R-BN provides a simple, data-independent masking strategy that naturally aligns with the spectral structure of audio signals, offering a robust and generalizable alternative to rigid masking schemes.

Table 4: **Comparison of self-supervised audio pretraining methods.** Our R-BN masking improves over AudioMAE and MaskSpec across all benchmarks. ∗ denotes results from our evaluation.

| Method | AS-20k | AS-2M | ESC-50 | SPC-2 |
|---|---|---|---|---|
| SS-AST | 31.0 | - | 88.8 | 98.0 |
| MaskSpec | 32.3 | 47.1 | 89.6 | 97.7 |
| MaskSpec + Ours | 33.4$^{(+1.1\%)}$ | **47.6**$^{(+0.5\%)}$ | 90.4$^{(+0.8\%)}$ | 98.2$^{(+0.5\%)}$ |
| MAE-AST | 30.6 | - | 90.0 | 97.9 |
| Audio-MAE∗ | 36.1 | 46.3 | 94.1 | 98.3 |
| Audio-MAE + Ours | **36.8**$^{(+0.7\%)}$ | 47.2$^{(+0.9\%)}$ | **94.6**$^{(+0.5\%)}$ | **98.7**$^{(+0.4\%)}$ |

Table 5: **Comparison on VGG-Sound with audio-only, video-only, and audio-visual inputs.** Our Green3D and R-BN masking improve performance across all settings. ∗ denotes results from our evaluation.

| Method | Audio | Video | Audio-Video |
|---|---|---|---|
| MBT | 52.3 | **51.2** | 64.1 |
| CAV-MAE∗ | 58.5 | 45.6 | 64.3 |
| CAV-MAE + Ours | **59.1**$^{(+0.6\%)}$ | 46.4$^{(+0.8\%)}$ | **64.9**$^{(+0.6\%)}$ |

## 4.3 RESULTS FOR AUDIO-VISUAL

**Setup.** We evaluate our proposed modality-specific masking within the CAV-MAE framework (Gong et al., 2022b). Random masking in the original model is replaced by **Green3D** masking for video frames and **Regularized Blue Noise (R-BN)** masking for audio spectrograms, while all other components remain unchanged. Following prior work, we adopt a ViT-B backbone, apply a masking ratio of 75% to both modalities, and pretrain on the large-scale **VGGSound** dataset (Chen et al., 2020), which consists of ∼200K audio-visual clips across 309 classes, including non-speech audio, with strong natural correspondence between sound and vision. Since the original AudioSet setup cannot be fully reproduced, we follow the authors' publicly released VGGSound pipeline and evaluate both models under identical conditions to ensure fair comparison. Pretraining runs for 25 epochs, after which the encoder is evaluated under unimodal (audio-only, video-only) and multimodal (audio–visual) settings. Full details are in Appendix A.14.

**Results.** Table 5 reports results on VGGSound. Incorporating Green3D for video and R-BN for audio consistently improves performance across all evaluation modes: audio-only (+0.6%), video-only (+0.8%), and audio–visual (+0.6%). The unimodal gains indicate that structured noise masking enhances modality-specific representation learning, while the multimodal gains highlight improved cross-modal alignment. As CAV-MAE (Gong et al., 2022b) employs random masking for both streams, these results demonstrate that simple, data-independent structured noise can serve as a plug-and-play replacement, strengthening both unimodal and joint representations without adding objectives or computational cost.

## 4.4 ABLATIONS

We ablate mask colors and types using smaller subsets of standard datasets: **mini-Kinetics** (25% of K400) and **mini-SSv2** (50% of SSv2) for video, and **AudioSet-20K (AS-20k)** and **ESC-50** for audio. Additional ablations and qualitative results are provided in Appendices A.1-A.9.

**Impact of color noise on video masking.** Table 6 reports performance and reconstruction loss for different color noise masks. **Green** masking achieves the best accuracy on both mini-Kinetics and mini-SSv2, striking a balance between reconstruction difficulty and solvability (loss = 0.60). In contrast, **Blue** lowers reconstruction loss too much (0.41), making the task overly easy and limiting representation quality, while **Red** noise imposes excessive difficulty (0.85), also reducing accuracy. This shows that Green provides a more effective inductive bias for video masking by respecting spatiotemporal coherence.

**Impact of color noise on audio masking.** For audio, the trend is reversed: among the color variants in Table 6, **Blue** yields the best accuracy on AS-20k and ESC-50 with a low reconstruction loss (0.45). **Green** provides no benefit and underperforms **Blue**, while **Red** makes the reconstruction task overly difficult (loss = 0.61), leading to degraded accuracy. This shows that for audio blue noise provides most effective inductive bias for masking.

**Blue vs R-BN.** Table 7 compares the vanilla Blue noise with our regularized blue noise **R-BN** for audio. Our proposed **R-BN** improves the performance on both downstream datasets compared to vanilla 2D blue noise masking. This validates our assumption that regularizing the vanilla blue noise to introduce more separability in visible patches for audio masking results in better representations..

**3D vs. 2D video masking** Table 7 compares 2D and 3D masking strategies for video. Applying 2D Green noise uniformly across frames ignores temporal structure and performs only marginally better than random tube masking. In contrast, our proposed **Green3D** masking, which enforces spatiotem-

Table 6: **Impact of color masks.** Left: 3D masks for video, where Green3D performs best. Right: 2D masks for audio, where Blue is strongest among other color variants.

Table 7: **Blue vs R-BN and Green3D vs Green2D.** For audio (top), R-BN beats vanilla Blue masking. For video (bottom), Green3D outperforms tube and Green 2D masking.

| Color Mask | $\mathcal{L}_{\text{recon}}$ | mini Kinetics | mini SSv2 | $\mathcal{L}_{\text{recon}}$ | AS-20k | ESC-50 |
|---|---|---|---|---|---|---|
| | | **Video** | | | **Audio** | |
| Random | 0.67 | 51.6 | 52.8 | 0.52 | 36.1 | 94.1 |
| Blue | 0.41 | 50.9 | 52.1 | 0.45 | 36.5 | 94.2 |
| Red | 0.85 | 51.0 | 52.3 | 0.61 | 35.5 | 92.6 |
| Green | 0.60 | 52.7 | 54.5 | 0.57 | 36.4 | 94.1 |

| Masking type | $\mathcal{L}_{\text{recon}}$ | AS-20k | ESC-50 |
|---|---|---|---|
| Blue | 0.45 | 36.5 | 94.2 |
| R-BN | 0.49 | 36.8 | 94.6 |

| Masking type | $\mathcal{L}_{\text{recon}}$ | mini-Kinetics | mini-SSv2 |
|---|---|---|---|
| Tube | 0.67 | 51.6 | 52.8 |
| Green2D | 0.73 | 51.9 | 52.9 |
| Green3D | 0.60 | 52.7 | 54.5 |

poral coherence, yields clear gains on both mini-Kinetics and mini-SSv2 with lower reconstruction loss. These results confirm that explicitly modeling temporal dependencies through 3D masking is crucial for robust video representation learning.

**Masking ratio ablation** We additionally examine the effect of masking ratio for both video and audio. VideoMAE typically peaks around 90% video masking and AudioMAE around 80% audio masking, and our Green3D and R-BN follow the same trends (gains from 80 to 90% for video and performance peaks near 80% for audio). Detailed results are provided in Appendix A.3 (Tables A.3 - A.4).

## 5 CONCLUSION

Self-supervised learning via masked modeling overlooks inherent structures within different data modalities by relying on random masking. In this work, we show that structured noise-based masking offers a simple yet effective alternative, naturally aligning with the spatial, temporal, and spectral characteristics of video and audio data. By leveraging color noise distributions, our approach introduces structured masking without requiring handcrafted heuristics or additional data. Consistent improvements in multiple benchmarks demonstrate that such modality-aware masking enhances representation learning without increasing computational costs. These findings reinforce a broader perspective: self-supervised masked modeling benefits not just from masking large portions of data, but from doing so in a way that respects the structure of the modality itself.

**Reproducibility Statement** Reproducibility is a key priority of this work. Upon acceptance, we will release the full codebase and pretrained model weights to enable full replication of our results, including the exact hyper-parameter settings reported in the paper. All experiments were conducted with fixed random seeds for PyTorch, CUDA, and Python to ensure deterministic behavior.

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

# A APPENDIX

The Appendix consists of the following sections: A.1 Sigma value ablations, A.3 Masking ratio ablations, A.5 Computational Efficiency of Structured Noise Masking, A.6 Convergence Analysis, A.8 Qualitative Results on Video Object Segmentation, A.9 Qualitative Reconstruction and A.10 Pseudo-code for Regularized Blue Noise generation, A.12 Training details of Green3D masking for Video, A.13 Training details of Regularized Blue Noise masking for Audio, A.14 Training details for Audio-Visual masking. A.16 provides details about usage of LLMs in this project.

## A.1 SIGMA VALUE ABLATIONS

The choice of $\sigma_1$ and $\sigma_2$ in Eq. 7 determines the spatial and temporal characteristics of Green3D noise, influencing how occlusions evolve across frames. Lower $\sigma_1$ values retain fine details, while higher $\sigma_2$ values remove high-frequency components, impacting motion continuity and spatial structure. To evaluate this effect, we analyze five different configurations:

- **Fixed values:**
    - **Variant-1:** $\sigma_1 = 0.5, \sigma_2 = 2$, enforcing a strong separation between high and low frequencies while capturing mid-scale structures.
    - **Variant-2:** $\sigma_1 = 1.5, \sigma_2 = 3$, shifting towards large-scale occlusions by increasing both $\sigma_1$ and $\sigma_2$.
- **Randomized selection:**
    - **Variant-3:** $\sigma_1$ is sampled from $[0.5, 1.5]$ and $\sigma_2$ from $[2, 3]$, introducing controlled variation while maintaining a mid-frequency emphasis.
    - **Variant-4:** A wider range with $\sigma_1 \sim U(0.2, 1.7)$ and $\sigma_2 \sim U(0.8, 2.3)$, allowing greater variability in occlusion structures.
    - **Variant-5:** $\sigma_1 \sim U(0.4, 1.5)$ and $\sigma_2 \sim U(1.4, 3)$, balancing structure and adaptability.

Table A.1: Ablation on $\sigma_1$ and $\sigma_2$ in Green 3D noise. Selecting $\sigma$ values from a controlled range (Variant-5) achieves the best performance, balancing spatial coherence and temporal smoothness.

| Variant | mini-Kinetics | mini-SSv2 |
|---|---|---|
| Variant-1 | 52.3 | 54.3 |
| Variant-2 | 52.1 | 53.3 |
| Variant-3 | 52.2 | 54.4 |
| Variant-4 | 51.8 | 54.3 |
| Variant-5 | 52.7 | 54.5 |

Results in Table A.1 show that Variant-1 performs well, but increasing both $\sigma_1$ and $\sigma_2$ in Variant-2 degrades performance, likely due to excessive smoothing that removes fine-grained occlusions. The randomized variants (Variants 3-5) introduce adaptability, reducing sensitivity to specific values. Among them, Variant-5 achieves the best performance across mini-Kinetics and mini-SSv2, suggesting that sampling from an intermediate range provides an optimal balance between spatial coherence and temporal smoothness.

These findings underscore the importance of properly tuning the spectral distribution of structured noise. A rigid selection limits adaptability, while excessive randomness results in suboptimal occlusions. By allowing controlled variation in $\sigma_1$ and $\sigma_2$, Variant-5 achieves diverse yet structured occlusions, leading us to adopt it as our final configuration for effective video masked modeling.

**Limitation.** While our ablations highlight the effectiveness of Variant-5 as a balanced configuration, we acknowledge that this choice may not represent the true optimal values of $\sigma_1$ and $\sigma_2$. Since our approach is data-independent by design, directly learning these parameters from data contradicts our objective. Nonetheless, more principled strategies for identifying optimal sigma ranges, beyond manual tuning, could further enhance mask diversity and effectiveness. Exploring such alternatives represents a promising direction for future work.

Table A.2: Impact of window size $\Delta$ on R-BN masking within AudioMAE. The default multi–scale configuration $(3, 5)$ performs best, while all settings outperform vanilla blue noise.

| Window sizes ($\Delta$) | AS-20k | ESC-50 |
|---|---|---|
| (1,3) | 36.3 | 93.8 |
| (3,5) | **36.8** | **94.6** |
| (7,9) | 36.1 | 93.7 |

Table A.3: Impact of masking ratio for VideoMAE (Green3D noise). The standard ratio of 90% yields the best performance.

| Masking ratio | L2-loss | mini-Kinetics | mini-SSv2 |
|---|---|---|---|
| 80% | 0.48 | 51.6 | 53.8 |
| 85% | 0.53 | 52.4 | 54.4 |
| 90% | 0.60 | 52.7 | 54.5 |

Table A.4: Impact of masking ratio for AudioMAE (Regularized Blue noise). The standard ratio of 80% performs optimally.

| Masking ratio | L2-loss | AS-20k | ESC-50 |
|---|---|---|---|
| 75% | 0.47 | 36.4 | 93.9 |
| 80% | 0.49 | 36.8 | 94.6 |
| 85% | 0.53 | 36.3 | 93.4 |

## A.2 ABLATION: SENSITIVITY TO R-BN WINDOW SIZE AND DIRECTIONAL WEIGHTS

To evaluate the sensitivity of **Regularized Blue Noise (R-BN)** to its hyperparameters, we analyze the effect of the window parameters $\Delta$ and the directional weighting vector $w$ used during mask generation.

In our implementation, the $\Delta$ parameters specify the spatial window sizes used to measure local patch clustering. For example, using (3,5) evaluates the regularization over both a $3 \times 3$ and a $5 \times 5$ neighborhood, encouraging patch separation at two spatial scales: very local structure ($\Delta = 3$) and a slightly broader context ($\Delta = 5$). This multi–scale formulation was used as the default setting in the main experiments.

To assess sensitivity, we varied $\Delta$ to smaller $(1, 3)$ and larger $(7, 9)$ window ranges. As shown in Table A.2, performance remains stable within a narrow band, with the default multi-scale configuration yielding the best results. We attribute this to $(3, 5)$ enforcing separation at multiple relevant time–frequency scales, while excessively small windows allow larger clusters, and large windows over-regularize the mask. Importantly, all configurations outperform vanilla 2D blue noise masking (Hinojosa et al. (2024)), indicating that R-BN is robust to reasonable choices of $\Delta$ and does not depend on precise tuning.

For the directional weights, we use the default asymmetric setting $w = [0.4, 0.4, 0.1, 0.1]$, which mildly emphasizes two primary directions while keeping the overall regularization close to isotropic. We also evaluated a fully isotropic weighting $(1, 1, 1, 1)$. This variant produced slightly more clustered masks and marginally lower performance, indicating that the asymmetric weighting provides a small but consistent benefit. However, our R-BN masking is not highly sensitive to this choice of $w$.

## A.3 MASKING RATIO ABLATIONS

Tables A.3 and A.4 provide a detailed analysis of the impact of masking ratios on performance for video (Green3D noise) and audio (Regularized Blue noise or R-BN) masking. We evaluate different masking ratios and observe that the previously established values of 90% for video Tong et al. (2022) and 80% for audio Huang et al. (2022) continue to yield the best results. For both modalities, increasing or decreasing the masking ratio leads to suboptimal performance, confirming that high masking rates effectively balance reconstruction difficulty and representation learning. These results

further reinforce that structured noise masking naturally aligns with the redundancy inherent in each modality, making it an efficient alternative to purely random masking without requiring additional tuning.

## A.4 GRID AND BLOCK MASKING ABLATIONS

While several prior works have explored structured, non-random masking strategies, such as grid or block patterns in MAE, SimMIM, and SimSIM, these approaches are generally reported to underperform random masking in masked modeling. To verify that the improvements observed with Green3D are not simply due to the use of any structured mask, we evaluated grid, block, and tube masking within the same VideoMAE training setup used in our main experiments.

Table A.5 shows that these fixed spatial patterns consistently lag behind both Tube masking and Green3D, with performance drops of up to 2.2% on mini-SSv2 and 1.7% on mini-Kinetics. These results reinforce the idea that trivial structured masks, such as grid and block patterns, do not improve masked modeling, which explains why random masking remains the standard in MAE-based methods. To our knowledge, our method is the first data-independent masking strategy to consistently outperform random masking across video, audio, and multimodal settings without added computation, reflecting a principled mask design for spatiotemporal and time-frequency domains rather than a simple 2D extension.

Table A.5: **Comparison of Green3D with common non-learned structured masking strategies.** Grid, block, and tube masks underperform Green3D, highlighting the importance of modality-aware mask design for video masked modeling.

| Masking type | mini-Kinetics | mini-SSv2 |
|---|---|---|
| Grid | 51.0 | 52.3 |
| Block | 51.1 | 52.5 |
| Tube | 51.6 | 52.8 |
| Green3D | 52.7 | 54.5 |

## A.5 COMPUTATIONAL EFFICIENCY OF STRUCTURED NOISE MASKING

To verify that our method introduces no additional computational overhead, we precompute all noise patterns offline and store them in memory before pretraining. Table A.6 reports the parameters, FLOPs, memory usage, and training time for VideoMAE with random masking and with our Green3D masking on K400 using a ViT-B backbone. The results show that both variants are effectively identical across all metrics: parameters (94.21M), FLOPs (∼21G), and memory (∼22GB) remain unchanged. Training time per epoch differs marginally (15:18 vs. 15:31), confirming that Green3D masking is computationally equivalent to random masking. These findings highlight that our structured noise masking provides consistent accuracy improvements without incurring any extra cost in training efficiency.

Table A.6: Computational efficiency of VideoMAE with random masking and with Green3D masking on K400 (ViT-B). Both variants show identical parameters, FLOPs, and memory usage, with only a marginal difference in training time per epoch.

| Model | Masking Strategy | Params (M) | Flops (G) | Mem (GB) | Epoch Time (mm:ss) |
|---|---|---|---|---|---|
| VideoMAE | Random | 94.21 | 21.28 | 22.62 | 15:18 |
| VideoMAE+ our masking | Green3D | 94.21 | 21.02 | 22.68 | 15:31 |

## A.6 CONVERGENCE ANALYSIS

We analyze the convergence behavior of Green3D masking compared to VideoMAE. Table A.7 reports top-1 accuracy on SSv2 when models are pretrained on K400 for different numbers of epochs.

Both methods show similar convergence speed, but Green3D consistently outperforms VideoMAE throughout training. For example, at 200 epochs, Green3D achieves +0.7% improvement, which increases to +1.2% at 600 epochs and remains stable at later stages. These results confirm that Green3D not only converges reliably but also yields consistent gains across the entire training trajectory.

Table A.7: Convergence comparison on SSv2 with K400 pretraining. Green3D achieves consistent accuracy gains over VideoMAE at all epochs, with improvements increasing from +0.7% at 200 epochs to +1.2% at 600 epochs, while maintaining a similar convergence speed.

| Method | 200 | 400 | 600 | 700 | 800 |
|---|---|---|---|---|---|
| VideoMAE | 66.71 | 67.76 | 68.15 | 68.51 | 68.53 |
| VideoMAE + our Green3D | 67.42 | 68.45 | 69.48 | 69.74 | 69.76 |

### A.7 MASK VARIANCE AND SEED STABILITY ANALYSIS

To evaluate whether the improvements introduced by our structured masks stem from meaningful inductive biases rather than stochastic variation in mask generation, we assess the stability of Green3D under different random seeds. In this experiment, we generate three distinct Green3D masks by sampling independent random seeds during mask construction, while keeping all training settings, architectures, and data identical.

As shown in Table A.8, downstream performance varies within a narrow range of $< 0.2\%$ absolute on both mini-Kinetics and mini-SSv2. This variance level is comparable to or lower than the seed-to-seed fluctuations typically reported for MAE-based pretraining under similar compute budgets (Tong et al., 2022; Hinojosa et al., 2024). Importantly, the gains introduced by Green3D over random masking and 2D colored masks (e.g., +1.7% on mini-SSv2; +1.1% on mini-Kinetics) are significantly larger than this variance margin, indicating that the improvements are systematic rather than noise-driven.

These findings confirm that the effectiveness of Green3D does not depend on a particular random instantiation of the mask, and that its performance benefits generalize across stochastic realizations.

### A.8 QUALITATIVE RESULTS ON VIDEO OBJECT SEGMENTATION

Figure A.1 shows qualitative examples on the DAVIS dataset, comparing segmentation masks from VideoMAE and VideoMAE+Green3D. Consistent with our quantitative results (Sec. 4.1.2), Green3D masking produces sharper, more object-centric segmentations and preserves temporal consistency across frames. These visualizations confirm that structured masking not only improves mIoU but also enhances the interpretability of learned spatiotemporal representations.

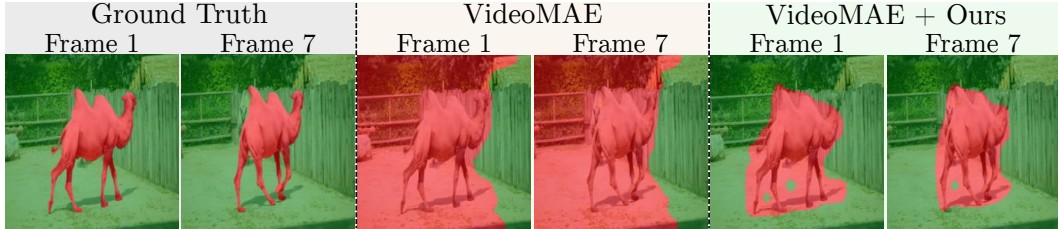

Figure A.1: **Qualitative results on DAVIS.** Green3D masking produces sharper and temporally consistent segmentations compared to VideoMAE.

### A.9 QUALITATIVE RECONSTRUCTION

To further analyze the impact of different masking strategies, we provide qualitative reconstruction results for video and audio masked modeling. Figures A.2 and A.3 compare VideoMAE pretraining

Table A.8: Performance variance across different Green3D mask realizations generated with independent random seeds. Variability remains below 0.2% absolute, indicating that improvements are not driven by stochastic noise patterns.

| Green3D mask (seed) | mini-SSv2 | mini-Kinetics |
|---|---|---|
| Seed A | 54.3 | 52.6 |
| Seed B | 54.5 | 52.8 |
| Seed C | 54.5 | 52.7 |
| **Mean** | **54.43** | **52.7** |

with different masking strategies on SSv2 at masking ratios of 0.75 and 0.9. Standard tube masking struggles to align with video structures, while 2D noise-based masking offers some spatial coherence but lacks temporal consistency. In contrast, our Green3D masking better captures spatiotemporal structures, preserving motion continuity across frames.

Figures A.4 and A.5 present spectrogram reconstructions for AudioMAE with a masking ratio of 0.8. Random masking results in scattered reconstructions, whereas red and green noise masking introduce artifacts that distort the frequency structure. Our Regularized Blue noise masking ensures a more balanced reconstruction by aligning with the spectral distribution of audio signals, demonstrating its effectiveness in preserving meaningful frequency patterns. These qualitative results further validate the advantages of our modality-aware structured noise masking in learning robust representations.

### A.10 PSEUDO-CODE FOR REGULARIZED BLUE NOISE GENERATION

In Algorithm 1, we present our Regularized Blue Noise masking strategy for audio pretraining. Unlike simple blue noise masking, our method explicitly enforces spatial separation between visible patches to ensure an appropriate distribution. Given a set of randomly ordered spatial positions, we iteratively assign visible patches by minimizing a clustering metric that evaluates local patch densities across multiple orientations. This optimization prevents undesirable patch clustering, leading to a more effective masking pattern for spectrogram-based representations.

### A.11 PSEUDO-CODE FOR MASKING FUNCTION

In Algorithm 2, we provide the pseudo-code of the masking function $\eta$. This function takes as input the sequence of token representations $X_p=\phi(X)$, a noise tensor $n$ (e.g., white, blue, green, or red noise), and a target mask ratio $\gamma$. It follows the standard procedure used in masked autoencoders: noise values are sorted and thresholded to select visible tokens, while the rest are masked. Our implementation is consistent with the functions adopted in the literature (He et al., 2022; Hinojosa et al., 2024).

### A.12 TRAINING DETAILS OF GREEN3D MASKING FOR VIDEO

**Pretraining details.** For VideoMAE (Tong et al., 2022) and SIGMA Salehi et al. (2024), we conduct pretraining on the Kinetics-400 (K400) (Kay et al., 2017) and Something-Something V2 (SSv2) (Goyal et al., 2017) datasets. We sample clips consisting of 16 frames at a spatial resolution of $224 \times 224$, applying temporal strides of 2 for SSv2 and 4 for K400. Each clip is processed into space-time tube embeddings using a 3D convolutional layer, with tokens defined by $2 \times 16 \times 16$ cubes. Pretraining is performed with an 90% masking ratio for 800 epochs, using 8 NVIDIA V100 GPUs. Additional configuration details are provided in Table A.9.

**Finetuning details for standard action recognition.** For full finetuning, we follow the protocol described by (Tong et al., 2022), utilizing 4 NVIDIA V100 GPUs. Complete finetuning settings are outlined in Table A.10.

**Unsupervised video object segmentation.** To conduct unsupervised segmentation evaluations, we extract video clips from the DAVIS Pont-Tuset et al. (2017) and YTVOS Xu et al. (2018) datasets. DAVIS Pont-Tuset et al. (2017) consists of 150 videos split into 60 for training, 30 for validation, and 60 for testing. Since the validation set is the only one that offers full-frame annotations, we use

---

**Algorithm 1:** Our Regularized Blue Noise Mask Generation

---

**Input:** Number of masks $K$, mask size $N_1 \times N_2$, window size $\Delta$, weights $w = [w_1, w_2, w_3, w_4]$,
      randomly ordered coordinates $\Omega$, transmittance ratio $\gamma$ ($0 < \gamma \leq 1$)
**Output:** Optimized masks $\hat{M}_b^0, \hat{M}_b^1, \ldots, \hat{M}_b^{K-1}$

1   Initialize $M^i \leftarrow \mathbf{0}_{N_1 \times N_2}$ for $i = 0, \ldots, K-1$;
2   Set maximum visible patches per mask: $V \leftarrow \gamma \times N_1 N_2$;
3   **for** *each spatial position* $(x, y)$ *in* $\Omega$ **do**
4      $\lambda \leftarrow \infty$, $\hat{i} \leftarrow -1$;
5      **for** $i = 0$ **to** $K - 1$ **do**
6          **if** $\sum(M^i) \geq V$ **then**
7              **continue**;
8          Extract local window $U_P^i$ of size $\Delta \times \Delta$ around patch $P = (x, y)$ from $M^i$.
9          Count patches:
10           $d_1^i \leftarrow$ horizontally from center $(x, y)$ in $U_P^i$;
11           $d_2^i \leftarrow$ vertically from center $(x, y)$ in $U_P^i$;
12           $d_3^i \leftarrow$ along main diagonal from center $(x, y)$ in $U_P^i$;
13           $d_4^i \leftarrow$ along second diagonal from center $(x, y)$ in $U_P^i$;
14          Compute clustering metric: $S_P^i \leftarrow w_1 d_1^i + w_2 d_2^i + w_3 d_3^i + w_4 d_4^i$;
15          **if** $S_P^i < \lambda$ **then**
16              $\lambda \leftarrow S_P^i$;
17              $\hat{i} \leftarrow i$;
18      Set mask values at $P = (x, y)$::;
19      **for** $i = 0$ **to** $K - 1$ **do**
20          **if** $i = \hat{i}$ **then**
21              $M_{x,y}^i \leftarrow 1$ `// Visible`
22          **else**
23              $M_{x,y}^i \leftarrow 0$ `// Masked`
24   **return** $M^0, M^1, \ldots, M^{K-1}$;

---

it to evaluate our segmentation performance. YTVOS Xu et al. (2018) is a larger dataset containing 4,453 videos across 65 categories. Ground truth masks are available only for the initial frames of test and validation videos. Consequently, we evaluate performance on a random 20% subset of the training set, ensuring consistent object class IDs using provided metadata.

We extract video clips from the DAVIS Pont-Tuset et al. (2017) and YTVOS Xu et al. (2018) using clip lengths of 16 frames and 4 frames, respectively. Each clip, along with its corresponding ground truth annotation, is passed through the encoder to obtain dense feature representations of dimensions $[\frac{T}{2}, d, 14, 14]$, with $d$ representing encoder dimensionality. Ground truth annotations and feature maps are resized to $28 \times 28$ resolution using nearest neighbor interpolation and linear interpolation methods, respectively. Clustering is performed with parameter $K$, aligned with the true object counts for standard clustering, and set three times higher for over-clustering scenarios. Clusters are subsequently duplicated and grouped to match ground-truth labels via either pixel-wise precision or the Hungarian matching method, as described by Salehi et al. (2023).

**Details of subsets and evaluation in SEVERE Benchmark** The SEVERE-Benchmark covers eight experimental settings built on four datasets, that is, Something-Something V2 Goyal et al. (2017), UCF (Soomro et al., 2012; Zhang et al., 2020), FineGYM Shao et al. (2020), and Charades (Sigurdsson et al., 2016). The detailed configuration of each subset is summarized in Table A.11. For finetuning and evaluation, we use the original codebase of the SEVERE benchmark Thoker et al. (2022), utilizing 4 NVIDIA V100 GPUs as shown in Table A.10.

### A.13   TRAINING DETAILS OF REGULARIZED BLUE NOISE MASKING FOR AUDIO

**Pretraining details.** For AudioMAE (Huang et al., 2022), we conduct pretraining on AudioSet-2M (AS-2M) (Gemmeke et al., 2017), following the original setup. Audio recordings are first transformed into 128-band log Mel spectrograms using a 25ms Hanning window with a 10ms hop size, resulting in spectrograms of size $1024 \times 128$ for 10-second clips. These spectrograms are parti-

**Algorithm 2:** Pseudo-Code of masking function ($\eta$) in PyTorch style.

```
1  import torch

2  def mask_generation(Xp, n, mask_ratio, T):
3      # Xp:  input tensor (image/video/audio) after patch embedding with φ; shape [B, N, d].
4      # n:  noise tensor ( white (n_w), blue(n_b), green(n_g), or red (n_r) noise).
5      # mask_ratio:  the mask ratio (γ) of total patches (e.g., 0.75).
6      # T: torchvision composed transform (random crop + horizontal flip + vertical flip).

7      # Infer batch size (B) and number of patches
8      B, N = Xp.shape[:2]
9      # apply random transforms (T)
10     windows = T(n)[:B] # Assuming B < n.shape[0]
11     len_keep = int(N * (1 - mask_ratio))
12     windows = windows.view(B, -1)

13     # keep stronger values from the noise
14     ids_shuffle = torch.argsort(windows, dim=1, descending=True)
15     ids_restore = torch.argsort(ids_shuffle, dim=1)
16     ids_keep = ids_shuffle[:, :len_keep]

17     # generate the binary mask:  1 = visible (keep), 0 = masked (remove)
18     mask = torch.zeros([B, N])
19     mask[:, :len_keep] = 1

20     # unshuffle to get the binary mask (M)
21     mask = torch.gather(mask, dim=1, index=ids_restore)
22     return mask, ids_restore, ids_keep
```

Table A.9: **VideoMAE and SIGMA pretraining setup.**

| config | SSv2 | K400 |
|---|---|---|
| optimizer | AdamW | |
| base learning rate | 1.5e-4 | |
| weight decay | 0.05 | |
| optimizer momentum | $\beta_1, \beta_2 = 0.9, 0.95$ | |
| batch size | 256 | |
| learning rate schedule | cosine decay | |
| warmup epochs | 40 | |
| flip augmentation | no | yes |
| augmentation | MultiScaleCrop | |

tioned into $16 \times 16$ non-overlapping patches, which are then linearly embedded and fed into the model. Pretraining uses an 80% masking ratio, in line with prior findings that high masking rates are effective for audio (Huang et al., 2022). The encoder consists of a 12-layer ViT-Base, while the decoder follows a 16-layer Transformer with local attention. Pretraining is performed for 32 epochs using 8 NVIDIA A5000 GPUs, a batch size of 512, and an AdamW optimizer with a base learning rate of 2e-4 and a cosine decay schedule.

**Finetuning details for audio classification.** For finetuning, we discard the decoder and fine-tune the ViT-B encoder with an additional classification head. The masking ratio is reduced to 30% (time-frequency masking) during fine-tuning, as lower masking improves classification performance (Huang et al., 2022). The model is optimized for 100 epochs on AS-2M and 60 epochs on AS-20K, using 8 NVIDIA A5000 GPUs. Fine-tuning follows a cosine decay learning rate schedule, starting at 1e-3, with an AdamW optimizer and a batch size of 256. For ESC-50, we adopt the standard 5-fold cross-validation protocol.

During evaluation on AudioSet, we use the standard test split, which contains approximately $20,000$ samples. However, due to copyright restrictions, YouTube periodically removes certain videos, resulting in variations in the exact test set used across different works. The original AudioMAE paper (Huang et al., 2022) did not release their exact test split for this reason. Instead, we use the publicly available AudioSet test set from Hugging Face, which contains a reduced number of samples compared to the original split. Significantly, we do not retrain AudioMAE but instead

Table A.10: **VideoMAE and SIGMA fine-tuning setup.**

| config | SSv2 | K400 | SEVERE |
|---|---|---|---|
| optimizer | | AdamW | |
| base learning rate | | 1.0e-3 | |
| weight decay | | 0.05 | |
| optimizer momentum | | $\beta_1, \beta_2 = 0.9, 0.999$ | |
| layer-wise lr decay(Bao et al., 2021) | | 0.75 | |
| batch size | 32 | 16 | 16 |
| learning rate schedule | | cosine decay | |
| warmup epochs | | 5 | |
| training epochs | 40 | 100 | 100 |
| flip augmentation | *no* | *yes* | *yes* |
| RandAug (Cubuk et al., 2020) | | (9,0.5) | |
| label smoothing(Szegedy et al., 2015) | | 0.1 | |
| mixup (Zhang et al., 2018) | | 0.8 | |
| cutmix (Yun et al., 2019) | | 1.0 | |
| drop path | | 0.1 | |

Table A.11: **Details of evaluation subsets in SEVERE benchmark (Thoker et al., 2022)**

| Evaluation Setup | Experiment | Dataset | Task | #Classes | #Finetuning | #Testing | Eval Metric |
|---|---|---|---|---|---|---|---|
| | Gym99 | FineGym | Action Class. | 99 | 20,484 | 8,521 | Top-1 Acc. |
| **Sample Efficiency** | UCF ($10^3$) | UCF 101 | Action Class. | 101 | 1,000 | 3,783 | Top-1 Acc. |
| | Gym ($10^3$) | FineGym | Action Class. | 99 | 1,000 | 8,521 | Top-1 Acc. |
| **Action Granularity** | FX-S1 | FineGym | Action Class. | 11 | 1,882 | 777 | Mean-per-class |
| | UB-S1 | FineGym | Action Class. | 15 | 3,511 | 1,471 | Mean-per-class |
| **Task Shift** | UCF-RC | UCFRep | Repetition Counting | - | 421 | 105 | Mean Error |
| | Charades | Charades | Multi-label Class. | 157 | 7,985 | 1,863 | mAP |

evaluate its publicly available pretrained checkpoints on our test set. This ensures a fair comparison, as both AudioMAE and our model are evaluated on the same dataset. While absolute numbers may differ slightly from those reported in (Huang et al., 2022), this discrepancy arises solely from variations in the available test data and does not affect the validity of our findings.

Table A.12: **AudioMAE pretraining setup.**

| Config | Value |
|---|---|
| Optimizer | AdamW |
| Base learning rate | 2e-4 |
| Weight decay | 0.05 |
| Optimizer momentum | $\beta_1, \beta_2 = 0.9, 0.95$ |
| Batch size | 512 |
| Learning rate schedule | Cosine decay |
| Warmup epochs | 5 |
| Training epochs | 32 |
| Masking ratio | 80% |
| Patch size | $16 \times 16$ |
| Encoder | ViT-Base (12 layers) |
| Decoder | Transformer (16 layers) |

## A.14 TRAINING DETAILS FOR AUDIO-VISUAL MASKING

**Pretraining details.** For CAV-MAE (Gong et al., 2022b), we pretrain on VGGSound (Chen et al., 2020), using 10-second audio-video clips. The audio spectrograms are computed using a 25ms Hanning window with a 10ms step size, producing 128 Mel frequency bins. Each spectrogram is divided into non-overlapping $16 \times 16$ patches, following the preprocessing of Audio Spectrogram Transformer (AST) (Gong et al., 2021). For video, we sample 10 RGB frames per clip at 1 FPS, resize them to $224 \times 224$, and split them into $16 \times 16$ patches, as in ViT (Dosovitskiy et al., 2020).

Table A.13: **AudioMAE fine-tuning setup.**

| Config | Value |
|---|---|
| Optimizer | AdamW |
| Base learning rate | 1e-3 |
| Weight decay | 0.05 |
| Optimizer momentum | $\beta_1, \beta_2 = 0.9, 0.999$ |
| Batch size | 256 |
| Learning rate schedule | Cosine decay |
| Warmup epochs | 5 |
| Training epochs | 100 (AS-2M), 60 (AS-20K) |
| Masking ratio | 30% (time-frequency) |
| Patch size | $16 \times 16$ |
| Encoder | ViT-Base (12 layers) |

Each modality is processed separately using modality-specific encoders. We employ an independent masking strategy per modality, applying Green noise masking for video and Blue noise masking for audio. Pretraining is conducted for 25 epochs using 8 NVIDIA A5000 GPUs, following the hyperparameters detailed in Table A.14.

**Finetuning details for classification.** For finetuning, we evaluate CAV-MAE representations on VGGSound for audio-only, video-only, and audio-video classification tasks. We retain the pretrained encoder and append a randomly initialized classification head. Training follows the same settings as (Gong et al., 2022b), using balanced sampling and augmentation strategies. The full finetuning setup is provided in Table A.15.

Unlike the original CAV-MAE paper, which reports results on the AudioSet audio-video dataset, we conduct all experiments on VGGSound. AudioSet is not publicly available in a downloadable format due to copyright restrictions, requiring users to manually retrieve videos from YouTube. However, our attempts to download the dataset were blocked due to IP restrictions, preventing us from reproducing their setup. Instead, we follow the authors' official repository, which provides a training script specifically for VGGSound, and train both the CAV-MAE baseline and our model accordingly. While this results in different absolute numbers from those reported in (Gong et al., 2022b), our setup ensures a fair comparison, as both methods are trained and evaluated under identical conditions on VGGSound.

Table A.14: **CAV-MAE pretraining setup.**

| Configuration | VGGSound |
|---|---|
| Optimizer | AdamW |
| Base learning rate | 1e-4 |
| Weight decay | 5e-7 |
| Optimizer momentum | $\beta_1, \beta_2 = 0.95, 0.999$ |
| Batch size | 120 |
| Learning rate schedule | Cosine decay |
| Warmup epochs | 2 |
| Training epochs | 25 |
| Audio input size | $1024 \times 128$ spectrogram |
| Video input size | $224 \times 224$ frames (10 fps) |
| Masking ratios | 75% (audio), 75% (video) |

## A.15 FREQUENCY-DOMAIN ANALYSIS OF MASKING STRATEGIES

To complement the qualitative comparisons in Fig. A.2–A.5 Figures A.2, A.3, A.4, and A.5, we provide a brief frequency-domain analysis for both audio spectrograms (Fig. A.6a and videos (Fig. A.6b). For this analysis, we sampled 256 audio clips from AS-20k and 256 video clips from Kinetics-400. Audio log-magnitude spectrograms were computed using a 1024-point FFT with a Hann window and 50% overlap. For the video clips, we extracted short segments and computed their 3D FFT magnitude. We applied the same FFT procedure to the different mask strategies. Fi-

Table A.15: **CAV-MAE fine-tuning.**

| Configuration | VGGSound |
|---|---|
| Optimizer | AdamW |
| Base learning rate | 1e-4 |
| Weight decay | 0.05 |
| Batch size | 48 |
| Learning rate schedule | Cosine decay |
| Warmup epochs | 2 |
| Training epochs | 10 |
| Mixup (Zhang et al., 2018) | 0.8 |
| Cutmix (Yun et al., 2019) | 1.0 |
| Drop path | 0.1 |
| Label smoothing (Szegedy et al., 2015) | 0.1 |

nally, we computed the radial power spectrum by grouping Fourier coefficients into radial-frequency bins and report both the power spectrum and the cumulative energy for each masking strategy.

**Audio.** As shown in Fig. A.6a (right), audio spectrograms concentrate more than 95% of their energy in the lowest radial-frequency bins. Low-frequency masks (Red) introduce large contiguous occlusions that remove entire harmonic trajectories (see Fig. A.4-A.5). In contrast, Blue masks produce fine-grained, high-frequency masking patterns that preserve the harmonic structure while achieving stronger downstream results.

**Video.** Video frames follow low-to-mid-frequency spatial statistics, dominated by edges, contours, and textures. As shown in Fig. A.6b (right), the Green3D mask has a mid-frequency clustered spectrum that closely aligns with the video distribution. This structure produces coherent masking blocks that encourage better modeling of spatial and temporal context.

### A.16 USE OF LARGE LANGUAGE MODELS

We use chatgpt (a large language model) to aid in polishing the writing of this submission. The model was employed solely for improving clarity and readability; all ideas, technical content, and conclusions are our own.

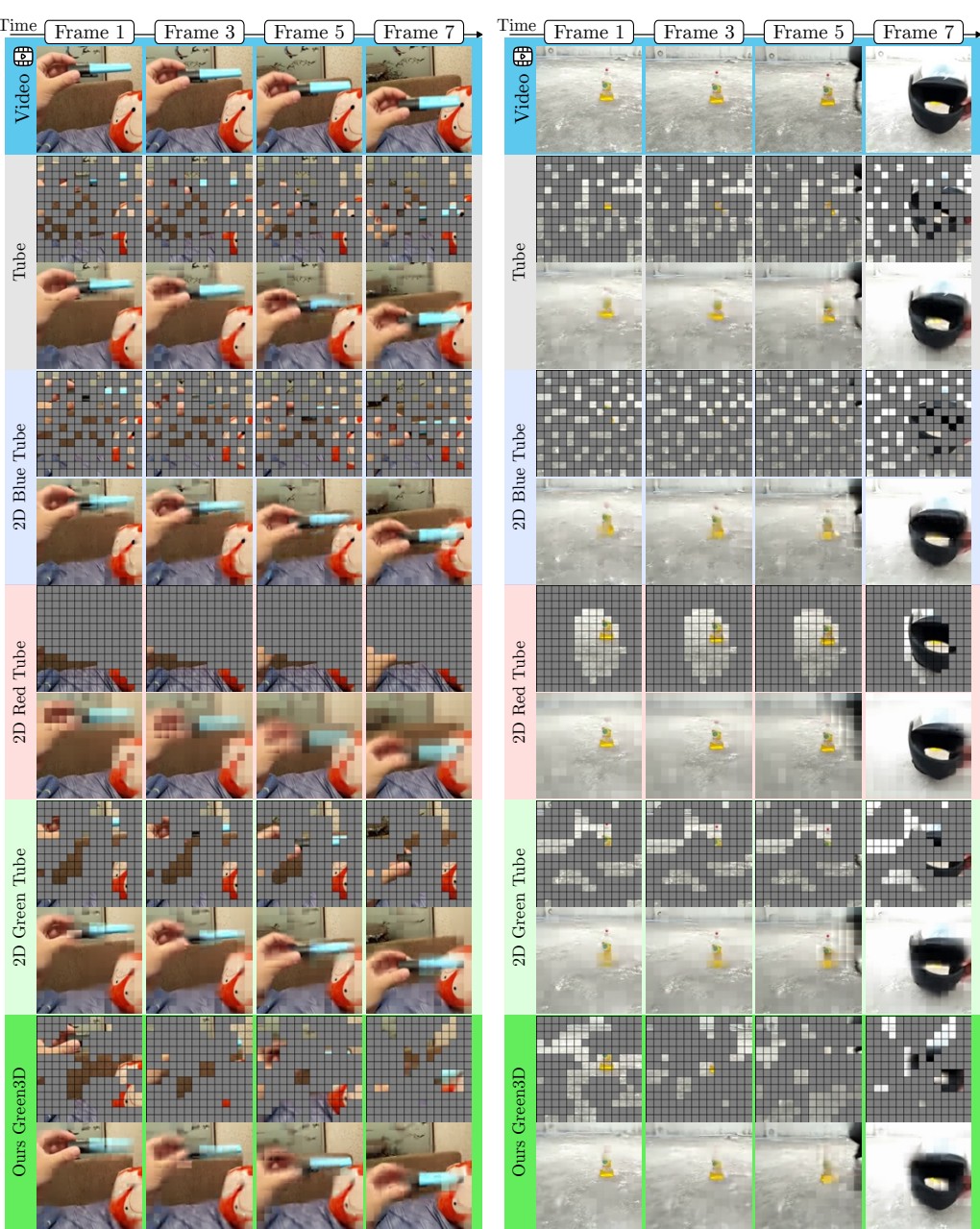

Figure A.2: Comparison of different masking strategies in VideoMAE pretraining on SSv2 videos (masking ratio 0.75). Standard tube masking struggles to align with video structures, while 2D noise-based masking introduces some spatial coherence but lacks temporal consistency. Our proposed Green3D masking effectively captures spatiotemporal structures, preserving motion continuity across frames.

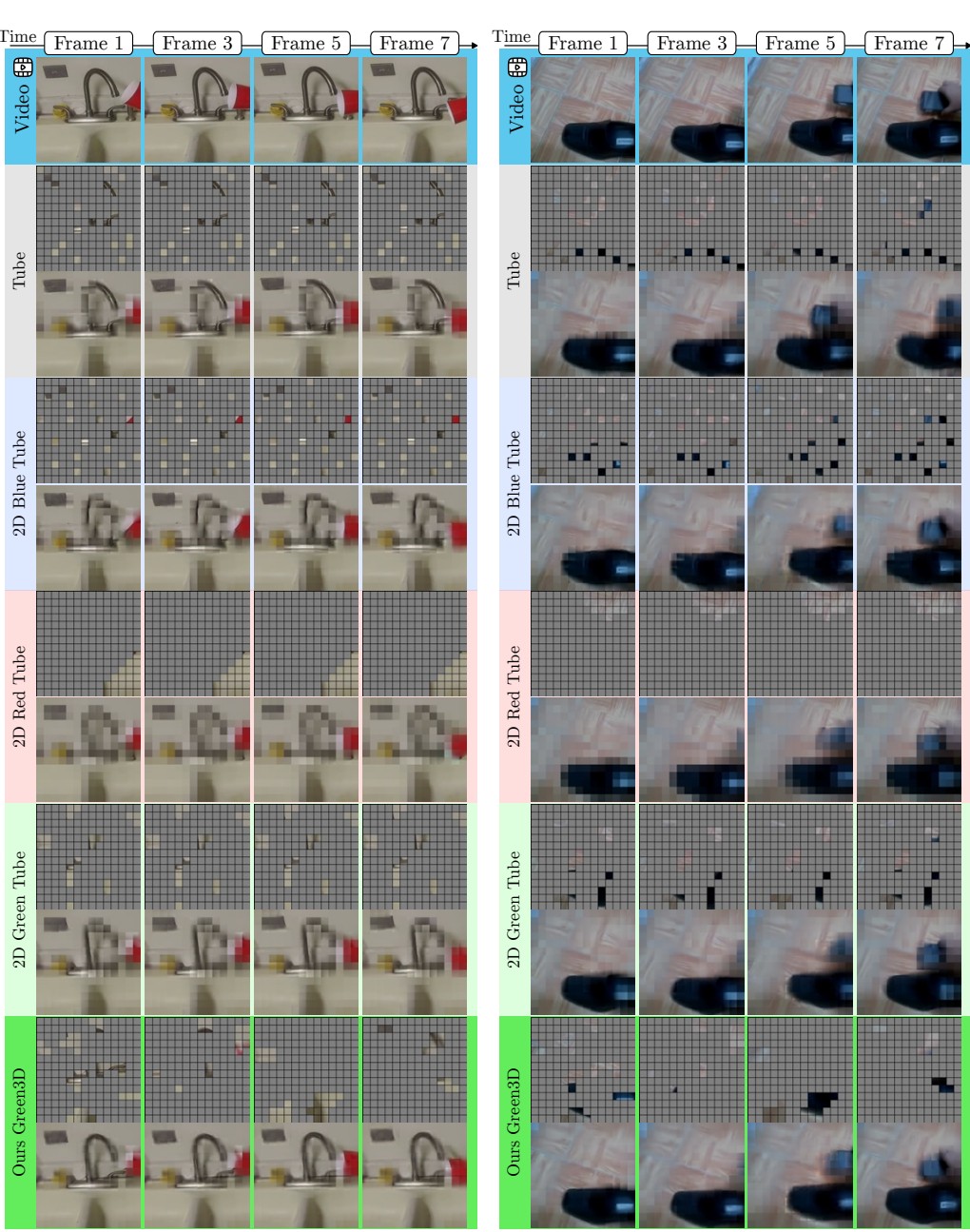

Figure A.3: Comparison of different masking strategies in VideoMAE pretraining on SSv2 videos (masking ratio 0.9). Standard tube masking struggles to align with video structures, while 2D noise-based masking introduces some spatial coherence but lacks temporal consistency. Our proposed Green3D masking effectively captures spatiotemporal structures, preserving motion continuity across frames.

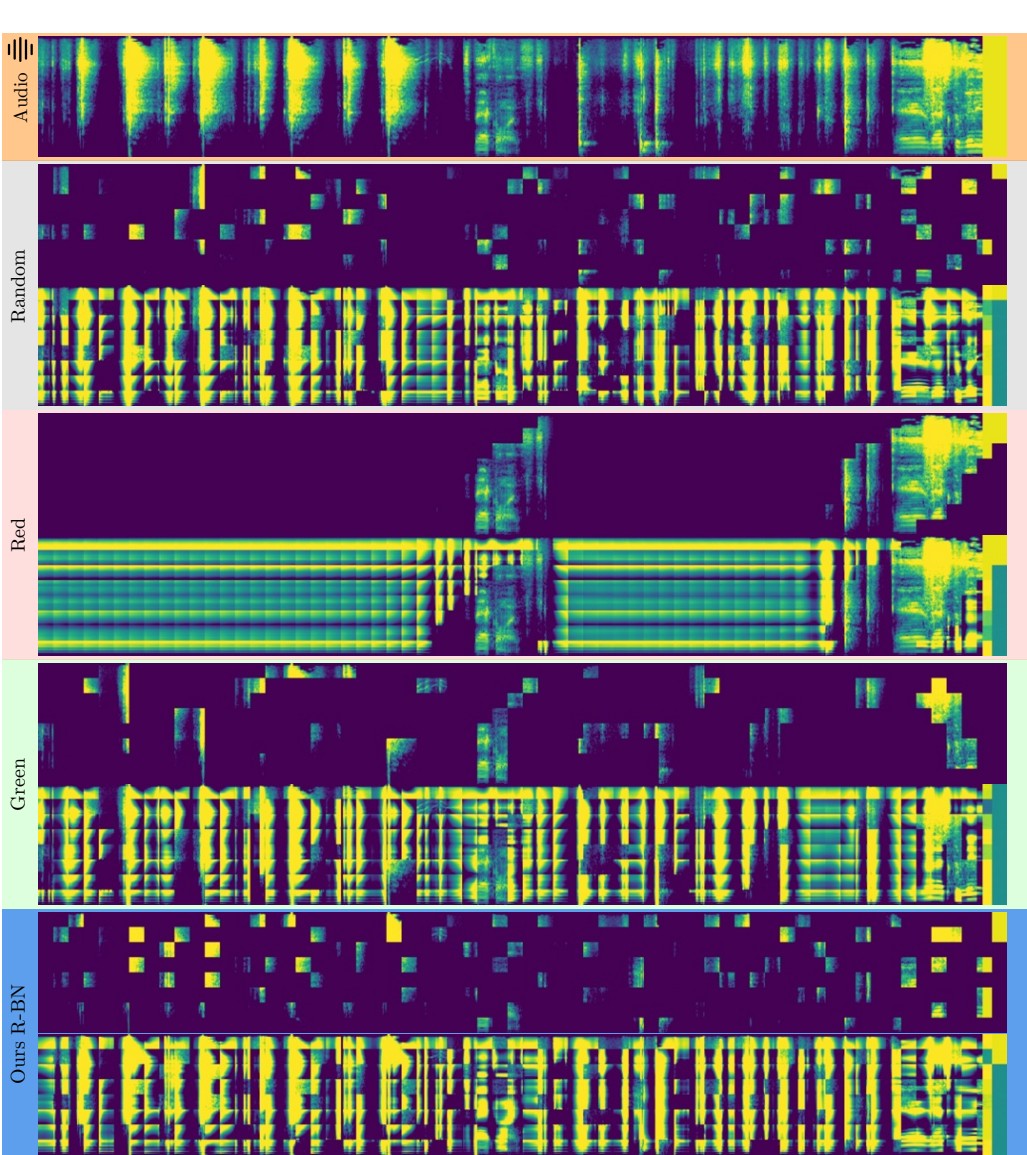

Figure A.4: Comparison of different masking strategies in AudioMAE pretraining on spectrograms (masking ratio 0.8). Random masking leads to scattered reconstructions, whereas red and green noise masking introduce biases that distort the frequency structure. Our proposed Regularized Blue noise masking ensures a more balanced reconstruction by aligning with the spectral distribution of audio signals.

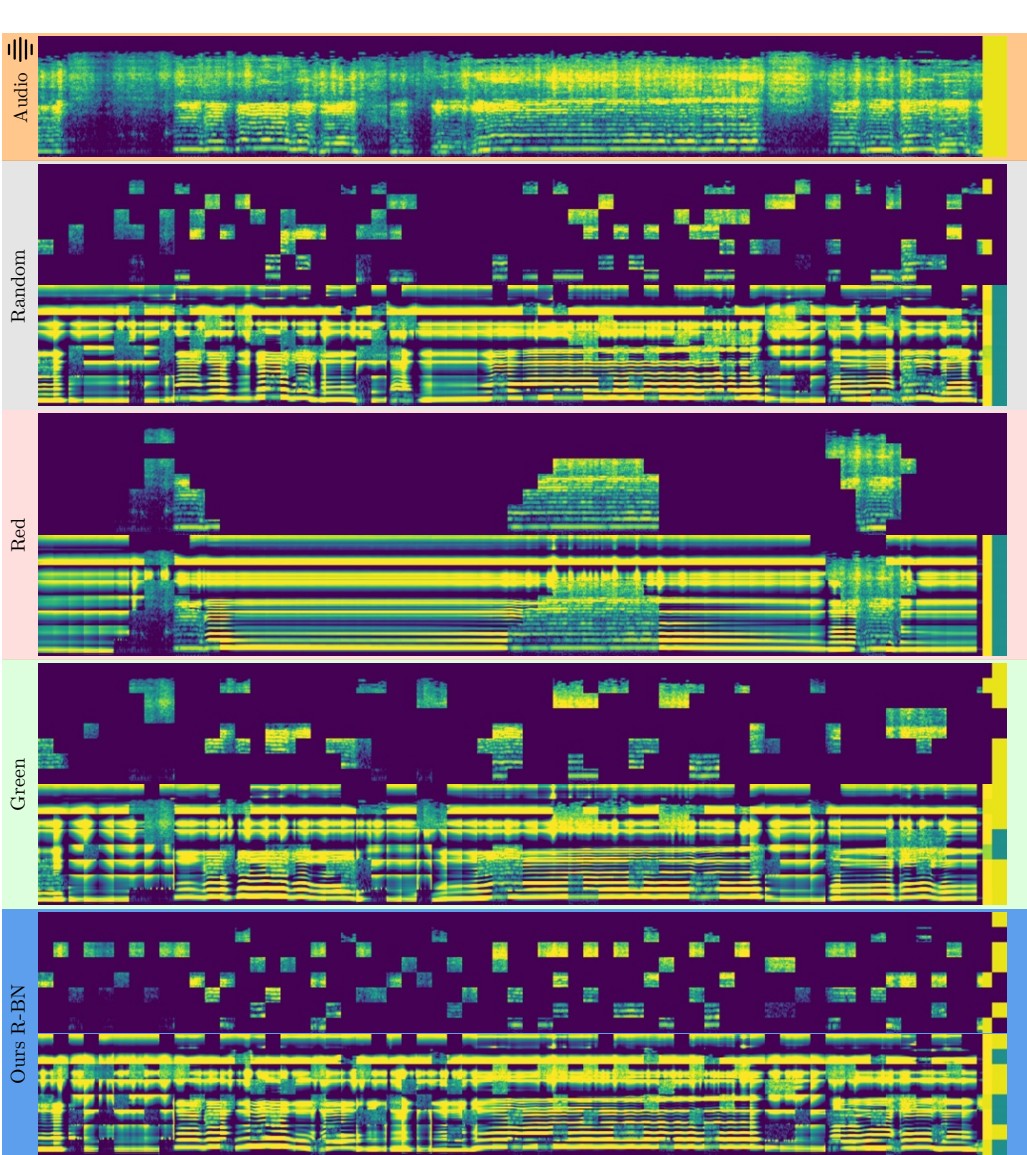

Figure A.5: Comparison of different masking strategies in AudioMAE pretraining on spectrograms (masking ratio 0.8). Random masking leads to scattered reconstructions, whereas red and green noise masking introduce biases that distort the frequency structure. Our proposed Regularized Blue noise masking ensures a more balanced reconstruction by aligning with the spectral distribution of audio signals.

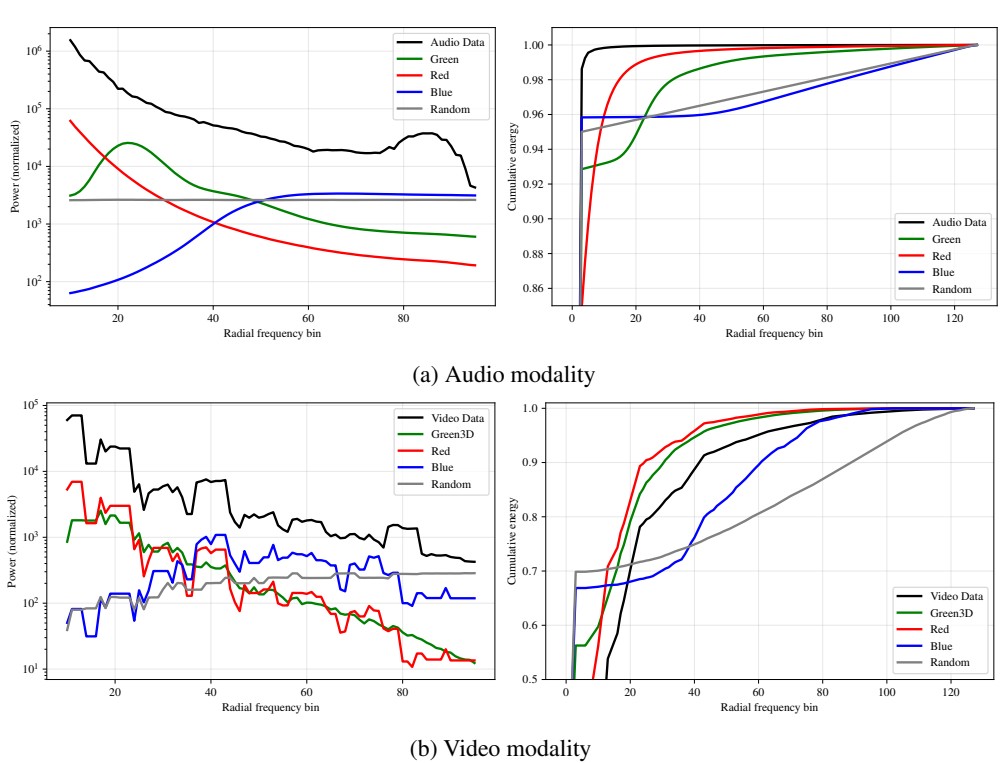

(a) Audio modality

(b) Video modality

Figure A.6: Frequency-domain analysis for (a) audio spectrograms and (b) video frames, comparing the modality spectrum with four masking strategies (Red, Green, Blue, Random).

