# OpenReview forum: "Structured-Noise Masked Modeling for Video, Audio and Beyond"
_ICLR.cc/2026/Conference — Submitted to ICLR 2026_

### Official Review · Reviewer_iQhj · 2025-10-29

**Soundness:** 2
**Presentation:** 3
**Contribution:** 2
**Rating:** 4
**Confidence:** 3

**Summary:**

The paper introduces a structured noise-based masking approach for self-supervised learning across multiple modalities such as video and audio. Its core contributions include: Green3D noise masking for video, which emphasizes spatio-temporal coherence; Regularized Blue Noise (R-BN) masking for audio, designed to ensure uniform distribution of visible patches across time and frequency; and the integration of both masking strategies in multimodal tasks to further enhance performance. The method is simple, incurs no additional computational cost, and achieves consistent improvements across a range of benchmarks, demonstrating both practical utility and general applicability.

**Strengths:**

1.The method is both simple and effective: it enhances self-supervised learning performance across multiple modalities using predefined noise structures, without relying on data or external supervision. All masks can be generated offline, introducing no additional computational burden during training.
2.Extensive experiments have been conducted: the approach has been thoroughly validated across video, audio, and multimodal tasks, including action recognition, video object segmentation, and audio classification, demonstrating its strong compatibility in multimodal settings.
3.The paper is well-structured and clearly articulated: it provides comprehensive background on color noise, detailed methodological descriptions, and systematic experimental validation, making the proposed approach easy to understand.

**Weaknesses:**

1. Lack of Theoretical and Intuitive Justification: The underlying rationale for the effectiveness of different color noises across modalities remains superficially explained. The claim that clustered masks "do not correspond to meaningful time-frequency events in audio" lacks sufficient theoretical support or empirical evidence from the audio domain. Similarly, while the advantage of Green noise for video is attributed to "spatiotemporal coherence," there is no in-depth analysis explaining why its mid-frequency clustering properties are superior to other structural patterns.
2. Insufficient Ablation Studies and Analysis:
    Hyperparameter Sensitivity: The selection of key hyperparameters (e.g., the sigma range for Green3D noise, the window size and weights for R-BN) appears empirical. A systematic ablation study investigating their sensitivity is lacking, making it difficult to assess the robustness and generalizability of the proposed method.
    Comparative Baselines: The ablation experiments primarily compare against random masking and other color noises. Comparisons with other simple, data-independent structured masking strategies (e.g., grid masking, block masking) are needed to better demonstrate the contribution of the proposed noise patterns.
    Performance Variance: It remains unclear whether the reported improvements consistently outweigh the performance fluctuations inherent in the stochastic nature of the noise generation process itself.

3. Unexplained Experimental Results: Only Table 3 do not indicate the performance delta of the proposed method. Certain metrics (e.g., UCF-RC) show a decrease after applying the proposed masking. The authors should discuss these instances to provide a more comprehensive perspective and clarify the potential limitations of their approach.

**Questions:**

1. Theoretical Motivation: Could you provide a more in-depth explanation from a signal processing perspective regarding why blue noise (emphasizing high-frequency, uniform distribution) is particularly suitable for audio spectrograms, while green noise (emphasizing mid-frequency clustering) is more appropriate for video? What fundamental structural differences between these modalities dictate this specific choice?
2. Hyperparameter Selection: How were the specific ranges for hyperparameters (e.g., σ₁ and σ₂ in Green3D, Δ and wᵢ in R-BN) determined? Was any systematic search or ablation study conducted? Could you discuss the sensitivity of model performance to variations in these parameters?
3. Completeness of Ablation Studies: Have you considered comparing your method with other non-random, data-independent masking strategies (such as grid or block masking)? Such comparisons would help strengthen the argument that the color noise structure itself is the key contributing factor to the performance gains.
4. Interpretation of Results: In Table 3, performance on the UCF-RC task decreases when using your method. What might be the potential reasons for this specific performance degradation? Does this suggest certain scenarios where your masking strategy might be less effective?
5. Performance Variance Analysis: The paper reports performance improvements across multiple benchmarks, but it remains unclear whether these gains consistently outweigh the performance fluctuations inherent in the stochastic noise generation process. Have you conducted statistical significance tests or multiple runs with different random seeds to verify that the improvements are statistically significant and not merely due to random variations in the noise patterns?
6. Lack of Citation: No citation or reference is provided for SPC-2 dataset in the paper. Please add the appropriate citation.

---

> ### Author Response · Authors · 2025-11-26
>
> We appreciate the insightful and constructive review, especially your recognition that our method is both "simple and effective," introduces "no additional computational burden," and is supported by "extensive experiments" across video, audio, and multimodal tasks. We address your remaining questions point-by-point below and provide deeper analysis and additional ablations where helpful.
>
> ------
>
> **W1/Q1 — Signal-Processing Motivation for Modality-Specific Masking**
>
> Thank you for the question. In addition to the qualitative pixel-domain comparisons provided in Fig. A.2-A.5, we have added a frequency-domain analysis for both audio spectrograms and videos in Fig. A.6 (a)  and Fig. A.6 (b), respectively, in our appendix.
>
> **Audio Modality:** As observed in Figs. A.4, A.5, and A.6(a), audio spectrograms concentrate energy in low-frequency harmonic bands due to the natural 1/f power distribution. Consequently, their semantic structure is highly sensitive to large occlusions that disrupt these harmonic trajectories. As shown in Fig. A.6 (a) (right), audio concentrates more than 95% of its energy in the first few radial-frequency bins. Any mask that carries low-frequency energy therefore removes broad and contiguous regions, which we visually confirm in Figs. A.4 and A.5, where Red masks erase whole harmonic stacks. For this reason, we argue that Blue (high-frequency) masking is particularly suitable for audio: it distributes removed patches uniformly and performs only fine-grained masking in the spectrogram domain, preserving the global harmonic structure. We further validate this effectiveness through extensive experiments in the paper.
>
> **Video Modality:** On the other hand, natural video frames exhibit low- to mid-frequency spatial statistics, with the majority of semantic structure appearing in this range, as observed in Fig. A.6(b). As shown in Fig. A.6(b) (right), the cumulative energy curve of Green3D is closest to that of the video data. Its mid-frequency clustered patterns, therefore, align with the spatiotemporal coherence of video.
>
> **Alignment with Prior Work:** This mask structure design also aligns with prior work findings. Specifically, [A] suggests that masking should target semantically informative regions while maintaining sufficient randomness to prevent trivial solutions. This is exactly what Green3D achieves: because it is generated from filtered noise, it remains stochastic; yet, its clustered mid-frequency structure creates contiguous masking blocks. This prevents the model from exploiting trivial local interpolation, thereby promoting the learning of more robust, global structures.
>
> In summary, the fundamental difference dictating this choice is that audio semantic structure relies on continuous harmonic trajectories (best preserved by fine-grained Blue masking), whereas video semantic structure relies on spatiotemporal object coherence (best targeted by clustered Green masking).
>
> [A] Haochen Wang, et al. "Hard patches mining for masked image modeling." *Proceedings of the IEEE/CVF Conference on Computer Vision and Pattern Recognition*. 2023.

---

> ### Author Response · Authors · 2025-11-26
>
> **W2/Q2 — Hyperparameter selection unclear, with no systematic analysis of σ-ranges or R-BN parameters and limited discussion of sensitivity.**
>
> Thank you for raising this question. We clarify how the key hyperparameters were chosen and summarize the sensitivity analyses we performed.
>
> **Green3D ($\sigma_1,\sigma_2$).** The $\sigma$ values control the spatial and temporal scale of the clusters formed in the mask. In practice, we first inspected the effect of different $\sigma$ values on the resulting 3D masks at the patch-grid resolution and discarded values where the masks became almost white noise (very small $\sigma$) or produced large blobs (very large $\sigma$). This led us to restrict both $\sigma_1$ and $\sigma_2$ to the bounded range [0.5, 2], yielding mid-scale spatiotemporal clusters spanning a few patches and frames. Within this interval, we ran a small grid of $\sigma_1 - \sigma_2$ combinations and included these experiments in Appendix A.1. The downstream accuracy varied only slightly across the tested settings, and all reasonable choices consistently outperformed both Random and 2D ColorMAE masks. These results suggest that Green3D is robust and does not require precise tuning of $\sigma_1$ and $\sigma_2$.
>
>
> **R-BN (Δ and directional weights).**
> In our implementation, the Δ parameters specify the spatial window sizes used to measure local patch clustering. For example, using [3,5] evaluates the regularization over both a 3\times3 and a 5\times5 neighborhood, encouraging patch separation at two spatial scales: very local structure (Δ=3) and a slightly broader context (Δ=5).
>
> To assess sensitivity, we varied Δ to smaller window sizes [1,3] and larger ones [7,9]. As shown below, performance remains stable across the tested settings, with the default multi-scale configuration providing the best results:
>
> | Window sizes (Δ) | AS-20k | ESC-50 |
> |------------------|--------|--------|
> | [1,3]            | 36.3   | 93.8   |
> | **[3,5]**        | **36.8** | **94.6** |
> | [7,9]            | 36.1   | 93.7   |
>
> We attribute this to the [3,5] configuration, which enforces separation across multiple relevant time-frequency scales, while very small windows provide only weak regularization (allowing clusters to form more easily), and large windows over-regularize the mask by suppressing fine-grained structure. Significantly, all configurations still outperform the underlying Blue mask, indicating that R-BN is robust to reasonable choices of Δ and does not require precise tuning.
>
> For the directional weights, we use the default asymmetric setting w = [0.4, 0.4, 0.1, 0.1], which mildly emphasizes two primary directions while keeping the overall regularization close to isotropic. We also evaluated a fully isotropic weighting (1, 1, 1, 1). This variant produced slightly more clustered masks and marginally lower performance, indicating that the asymmetric weighting provides a small but consistent benefit, although the method is not highly sensitive to this choice. We have added this discussion to Appendix A.2 in the revised manuscript.
>
> **Masking ratio sweeps.**
> We already report masking-ratio ablations for video (Green3D) and audio (R-BN) in Appendix A.3 (Tables A.3–A.4), and we have made these results more visible in the revised manuscript. For convenience, we also include the results below. They mirror the established behavior in prior work: VideoMAE typically peaks around 90% video masking (Tong et al., 2022) and AudioMAE around 80% audio masking (Huang et al., 2022), and our Green3D and R-BN follow the same trends (gains from 80 to 90% for video and performance peaks near 80% for audio). We have added this ablation in the revised version of the manuscript.
>
> **Masking ratio ablation for Green3D masking on VideoMAE**
>
> | Masking Ratio | L2-loss | mini-Kinetics | mini-SSv2 |
> |---------------|---------|---------------|-----------|
> | 80%           | 0.48    | 51.6          | 53.8      |
> | 85%           | 0.53    | 52.4          | 54.4      |
> | **90%**       | 0.60  | **52.7**      | **54.5**  |
>
>
> **Masking ratio ablation for R-BN masking on AudioMAE**
>
> | Masking Ratio | L2-loss | AS-20k | ESC-50 |
> |---------------|---------|--------|--------|
> | 75%           | 0.47    | 36.4   | 93.9   |
> | **80%**       | 0.49  | **36.8**| **94.6** |
> | 85%           | 0.53    | 36.3   | 93.4   |

---

> ### Author Response · Authors · 2025-11-26
>
> **W2/Q3 — Missing comparisons to other simple, non-random masking strategies (e.g., grid/block), limiting clarity on whether gains stem from color-noise structure itself.**
>
> Following the reviewer's suggestion, we explicitly evaluated grid and block masking within the VideoMAE training pipeline under the same settings used for our Green3D ablations, and we present the results in the table below.
>
> | Masking Type | mini-SSv2 | mini-Kinetics |
> |--------------|-----------|---------------|
> | Grid         | 52.3      | 51.0          |
> | Block        | 52.5      | 51.1          |
> | Tube         | 52.8      | 51.6          |
> | Green3D      | 54.5      | 52.7          |
>
> These experiments confirm observations reported in prior masked-modeling literature (e.g., MAE, SimMIM, SimSIM): structured masks with fixed spatial layouts (e.g., grids or contiguous blocks) consistently underperform random masking.
>
> In practice, we found that:
>
> - Grid masking leads to overly regular, low-frequency occlusions that make reconstruction too easy and reduce the diversity of visible regions.
>
> - Block masking removes large contiguous areas that rarely align with natural motion boundaries, degrading spatiotemporal learning.
>
> Both strategies yield lower downstream accuracy than random Tube masking and are considerably below Green3D. These numbers reinforce that the performance gains arise from the modality-aligned frequency structure of Green3D rather than from the use of any structured mask. We have added these additional comparisons in Appendix A.4 of the revised manuscript.
>
> ------
>
> **W3/Q4 — Performance on UCF-RC drops, raising questions about why degradation occurs and whether the masking strategy is less effective in certain scenarios.**
>
>
> Thank you for pointing this out. We agree that the direction of the UCF-RC metric may not have been immediately clear in the table, and we provide a clarification below.
>
> UCF-RC reports mean error in repetition counting (Zhang et al., 2020), where lower values indicate better performance. Because this metric is inverted relative to the classification metrics in the same table, the improvement can be easily misinterpreted at a glance. Under this metric, our Green3D masking achieves a lower repetition-counting error than the corresponding baseline, indicating a performance gain rather than a drop. We have made this directionality explicit in section 4.1.3 in the revised version to avoid any ambiguity.
>
> More broadly, UCF-RC is also one of the few tasks in SEVERE that measures cycle-accurate temporal reasoning. This task is susceptible to subtle temporal distortions. We observe that our method either maintains or improves performance on UCF-RC depending on the architecture and masking ratio used. This behavior is consistent with the improvements we see across all other temporal benchmarks, such as SSv2, Gym99, UCF-101, YTVOS, DAVIS, and the remainder of the SEVERE suite.
>
> We appreciate the opportunity to clarify this point and have updated the manuscript to present the metric definition and direction more clearly.

---

> ### Author Response · Authors · 2025-11-26
>
> **W2/Q5 — Unclear whether improvements are robust to stochastic variation; lacking multiple runs or significance analysis to rule out random noise effects.**
>
> Thank you for raising this question. We conducted an additional variance analysis to verify that the improvements are not due to stochastic fluctuations in mask generation.
>
> For Green3D, we generated three masking configurations with independent random seeds and trained separate VideoMAE models under identical settings. As shown below, the results exhibit very low variance (<0.2% absolute), despite the masks being structurally different:
>
> | Green3D variant | mini-SSv2 | mini-Kinetics |
> |-----------------|-----------|---------------|
> | Seed A          | 54.3      | 52.6          |
> | Seed B          | 54.5      | 52.8          |
> | Seed C          | 54.5      | 52.7          |
> | **Mean**        | **54.43** | **52.7**      |
>
> This variance level matches the seed variability typically reported for MAE-style models under similar training budgets. Importantly, the gains provided by Green3D over random masking and 2D colored masks (e.g., +1.7% on mini-SSv2, +1.1% on mini-Kinetics) are substantially larger than the observed variance, indicating that the improvements are systematic rather than noise-driven. We have included this ablation in Appendix A.7 of the revised manuscript.
>
> -----
>
> **Q6 – Lack of Citation: No citation or reference is provided for SPC-2 dataset in the paper. Please add the appropriate citation.**
>
> Thank you for pointing this out. We have added the appropriate citation for the SPC-2 dataset in the revised manuscript. The omission was unintentional, and we appreciate the reviewer bringing it to our attention.
>
> -------
>
> Thank you again for the detailed and constructive feedback. We have added a frequency-domain analysis, hyperparameter sensitivity studies, grid/block baselines, performance-variance results across seeds, clarification of the UCF-RC metric direction, and the missing SPC-2 citation. We hope these additions fully address your concerns about justification, robustness, and completeness. We would be sincerely grateful if you would consider an updated evaluation should you feel the revisions satisfactorily resolve your comments. Please let us know if you have any further questions or concerns; we would be happy to continue the discussion.

---

### Official Review · Reviewer_1dtC · 2025-10-30

**Soundness:** 3
**Presentation:** 3
**Contribution:** 2
**Rating:** 4
**Confidence:** 2

**Summary:**

The paper introduces a structured noise-based masking approach for video and audio masked modeling. In video masked modeling, the paper proposes 3D Green masking (Green3D)  using  3D Gaussian kernels for mask generation. In audio masked modeling, the paper proposes Regularized Blue Noise (R-BN) Mask Generation and calculates clustering score with four direction for mask generation. The experiments show good results.

**Strengths:**

1. The paper organization is easy to follow.
2. The mask modeling methods are understandable.
3. The experiment results are impressive.

**Weaknesses:**

1. Limited novelty:

This paper seems to apply the ColorMAE approach to audio and video tasks. The authors should emphasize what specific challenges arise when using a color-based MAE for audio and video, and how these challenges are addressed. Or the difference between the works and ColorMAE. In addition, an ablation study comparing ColorMAE with the proposed method should be provided.

2. Some details are unclear:

It is not explained why the masking strategy in Figure 3(c) performs better than that in Figure 3(b). The paper only presents the equations but lacks qualitative analysis to explain why it works better.

**Questions:**

Please see the weaknesses.
I will increase my rating if weakness 1 is solved.

---

> ### Author Response · Authors · 2025-11-26
>
> We appreciate the reviewer's thoughtful and constructive review and are grateful for your positive comments, noting that our paper is “easy to follow,” the proposed masking methods are “understandable,” and the experimental results are “impressive.” We address the remaining concerns point by point below and provide additional comparisons and explanations where helpful.
>
> -----------------
>
> **W1 — Limited novelty: viewed as applying ColorMAE to audio/video without clearly addressing modality-specific challenges or comparing against ColorMAE.**
>
> Thank you for your constructive feedback. We agree that our method is inspired by ColorMAE (Hinojosa et al., 2024), which obtains 2D masks by filtering noise, but its design is limited to 2D spatial structure. However, directly extending these 2D colored masks to video and audio is sub-optimal in practice, as we empirically validated in Table 7 of the main paper.
>
> When we apply the 2D Green mask from ColorMAE to the video data by repeating it across frames (Green2D), performance decreases. Likewise, applying the 2D Blue mask to audio spectrograms also underperforms. This is reflected clearly in the results below, where direct extensions of ColorMAE underperform our modality-aware designs under identical training settings:
>
> **Green2D mask from ColorMAE vs our Green3D applied on VideoMAE**
>
> | Masking Type | mini-SSv2 | mini-Kinetics |
> |--------------|-----------|---------------|
> | Green2D      | 52.9      | 51.9          |
> | Green3D      | 54.5      | 52.7          |
>
> **Blue mask from ColorMAE vs our R-BN applied on AudioMAE**
>
> | Masking Type | AS-20k | ESC-50 |
> |--------------|--------|--------|
> | Blue         | 36.5   | 94.2   |
> | R-BN         | 36.8   | 94.6   |
>
> These results demonstrate that a straightforward extension of ColorMAE is insufficient. Video requires spatiotemporal coherence: simply repeating a 2D mask or sampling it independently introduces temporal flicker or oversmoothing. Audio spectrograms are highly anisotropic, with harmonics along frequency and transients along time; 2D colored masks remove entire harmonic trajectories or cluster locally, disrupting meaningful time–frequency events.
>
> Motivated by these structural differences, our approach focuses on modality-aware mask design rather than a direct extension of ColorMAE, ensuring that the mask structure matches each modality's inherent properties. Additionally, in Appendix section A.15, we provide a frequency-domain analysis supporting this intuition, showing that Green3D matches the mid-frequency statistics of natural videos, while R-BN preserves harmonic structure in audio. We have clarified this motivation behind our mask design choice in section 3.3 of the revised manuscript.
>
> ------
>
> **W2 — Qualitative Analysis of Masking Performance in Fig. 3 (Random vs. Blue)**
>
> We apologize for the ambiguity in Figure 3. To clarify, Fig. 3(b) represents the initialization of our algorithm (which corresponds to standard Random Masking), while Fig. 3(c) represents the final output (our proposed R-BN Masking) after the optimization process.
>
> Regarding the qualitative analysis of why (c) performs better than (b): Fig. 3(b) shows that Random Masking produces stochastic clusters of masked patches. In the context of audio spectrograms, semantic structure is highly sensitive to such large occlusions because they can disrupt entire harmonic trajectories. In contrast, our R-BN (Fig. 3c) creates a high-frequency profile with uniformly distributed masked patches. This ensures that masked regions are always surrounded by visible patches, thereby preserving the global harmonic structure of audio.
>
> We validate this in our supplementary material:
>
> - Reconstruction: Figs. A.4 and A.5 confirm that Random Masking (Fig. 3b) produces localized clusters of masked patches that lead to truncated harmonic structures, whereas R-BN (Fig. 3c) reconstructs harmonics more consistently and preserves their continuity.
>
> - Frequency Analysis: Fig. A.6(a) shows that the high-frequency profile of R-BN complements the low-frequency nature of audio signals (1/f distribution), ensuring that the mask does not overlap structurally with the signal features.
>
> -------
>
> We appreciate the insightful and constructive review. We have clarified the differences from ColorMAE by detailing the modality-specific challenges in video and audio, added explicit comparisons showing why naïve ColorMAE extensions underperform, and provided clearer intuition for the masking strategy in Fig. 3(c). We hope these additions fully address your key concerns, and we would be grateful for your consideration of an updated evaluation if you feel the revisions satisfactorily resolve them.

---

### Official Review · Reviewer_KjQ5 · 2025-10-31

**Soundness:** 4
**Presentation:** 4
**Contribution:** 3
**Rating:** 8
**Confidence:** 3

**Summary:**

This paper proposes an alternative approach to random(or adoptive) masking socially for modalities where spatial and spatio-temporal continuity.  This paper extends structured noise masking to video by introducing 3D Green masking, a color noise–based strategy that produces spatiotemporally coherent masks, preserving spatial clustering and temporal smoothness. It introduces Regularized Blue noise for Audio, a 2D blue-noise-based optimization leveraging the structure of the audio. Standard methods, rely on random tube masking, which applies a static mask across frames, preserving temporal consistency but lacking adaptability to motion. The proposed  Green3D Noise Masking introduces structured, evolving masks across frames, enhancing fine-grained temporal representation learning.

**Strengths:**

Overall, this paper has a solid contribution towards better masking for Video and audio. The authors have presented results with strong baselines showing improvements on standard benchmarks, which is appreciated. Also the data independent approach of Green 3D makes it much more efficient than alternative approaches that required motion prior.  In terms of novelty,  this is an extension of image to video- this seems a bit incremental although important as it shows solid empirical evidence over baseline.  The results are particularly impressive for SEVERE benchmark, showing the generalization.

**Weaknesses:**

The paper contribution comes across as a bit incremental.
For the experiments, the following should be addressed.
1. Not sure if I followed the argument that this masking approach outperforms MGM in the in-domain setting and MGMAE in the cross-domain setting for SSv2. The comparison seems different as SIGMA is higher for Action Recognition
2. It would be great to see some ablation study done varying target masking ratio using these new masking approaches.


Minor comments:
1. It would be good to have a discussion for  alternative approaches for joint audio-visual masking

**Questions:**

1. Not sure if I followed the argument that this masking approach outperforms MGM in the in-domain setting and MGMAE in the cross-domain setting for SSv2. The comparison seems different as SIGMA is higher for Action Recognition
2. It would be great to see some ablation study done varying target masking ratio using these new masking approaches.

---

> ### Author Response · Authors · 2025-11-26
>
> Thank you for the thoughtful review. We sincerely appreciate your positive evaluation of our work, including your recognition of the "solid contribution toward better masking for video and audio," the "strong baselines showing improvements on standard benchmarks," and the efficiency of our "data-independent" Green3D design compared to motion-based approaches. We address your remaining comments point by point below, providing clarifications and additional results where helpful.
>
> ----------
>
> **W1/Q1 — Unclear comparison against MGM/MGMAE on SSv2, especially given SIGMA’s higher action recognition results.**
>
> We would like to clarify that we  are referring to the comparison between our masking with VideoMAE architecture (VideoMAE+Green3D) and motion-guided approaches, MGM and MGMAE, as follows:
>
> In the in-domain setting (pretraining on SSv2 followed by fine-tuning on SSv2), VideoMAE+Green3D attains 70.8% Top-1, slightly exceeding MGM at 70.6%. In the cross-domain setting (pretraining on Kinetics-400 then fine-tuning on SSv2), VideoMAE+Green3D reaches 69.7% Top-1, outperforming MGMAE at 68.9%.
>
> The key takeaway is that our data-independent masking is on par with, or better than, data-dependent masking approaches for SSv2 action recognition within the VideoMAE architecture and can further enhance more advanced architectures like SIGMA, without incurring any computational overhead.
>
> -----------
>
> **W2/Q2 — It would be great to see some ablation study done varying the target masking ratio using these new masking approaches.**
>
> Thank you for the suggestion. We already include masking-ratio ablations for video (Green3D) and audio (R-BN) in Appendix A.3 (Tables A.3–A.4), and we made these results more visible in the revised Appendix document. For convenience, we also include the results below. They mirror the established behavior in prior work: VideoMAE typically peaks around 90% video masking (Tong et al., 2022) and AudioMAE around 80% audio masking (Huang et al., 2022), and our Green3D and R-BN follow the same trends (gains from 80 to 90% for video and performance peaks near 80% for audio). We have added this ablation in section 4.4 of the revised version of the manuscript.
>
> **Masking ratio ablation for Green3D masking on VideoMAE**
>
> | Masking Ratio | L2-loss | mini-Kinetics | mini-SSv2 |
> |---------------|---------|---------------|-----------|
> | 80%           | 0.48    | 51.6          | 53.8      |
> | 85%           | 0.53    | 52.4          | 54.4      |
> | **90%**       | 0.60  | **52.7**      | **54.5**  |
>
>
> **Masking ratio ablation for R-BN masking on AudioMAE**
>
> | Masking Ratio | L2-loss | AS-20k | ESC-50 |
> |---------------|---------|--------|--------|
> | 75%           | 0.47    | 36.4   | 93.9   |
> | **80%**       | 0.49  | **36.8**| **94.6** |
> | 85%           | 0.53    | 36.3   | 93.4   |
>
> -----------
>
> **Minor Weakness — It would be good to have a discussion for alternative approaches for joint audio-visual masking**
>
> Thank you for raising this point. We clarify that most existing audio–visual masking strategies are learned or cross-modal, rather than predefined or data-independent. Prior works such as:
>
> - **MST: Masked Self-Supervised Transformer for Visual Representation** (Li et al, 2021) (Tube masking + audio noise), uses synchronized tube masks across video and audio channels, but does not incorporate modality-adaptive frequency properties;
>
> - **MultiMAE: Multi-modal Multi-task Masked Autoencoders** (Bachmann et al., 2022) extends masked autoencoders to multiple modalities by using shared latent tokens and cross-attention to jointly reconstruct images and other modalities. Masking is learned through the model's latent representations rather than predefined patterns, meaning it does not use fixed, data-independent masks.
>
> All of these approaches require either learned attention, joint encoders, or cross-modal conditioning. None provides a data-independent, plug-and-play masking strategy that works across video, audio, and audio-visual pipelines without modifying the architecture.
>
> In contrast, our method explicitly targets modality-aware mask design, and we show that our proposed masks (Green3D for video and R-BN for audio) already produce consistent gains in multimodal training using the unchanged CAV-MAE framework and identical hyperparameters. This suggests that better modality-aware masking can directly enhance audio-visual models without requiring cross-modal mask design.
>
> We agree that learned or cross-modal joint masking is a promising orthogonal direction, and we have added a brief discussion of these related approaches in section 2 of the revised manuscript.
>
> --------
>
> Thank you again for the positive and supportive feedback. We truly appreciate your careful reading of our work, and we hope our additional analyses and clarifications fully address your remaining comments. Please let us know if you have any further questions or concerns; we would be happy to continue the discussion.

---

### Official Review · Reviewer_NX9G · 2025-11-04

**Soundness:** 3
**Presentation:** 3
**Contribution:** 3
**Rating:** 6
**Confidence:** 4

**Summary:**

This paper introduces a data-independent, modality-aware masking approach for masked modeling. For video, it proposes Green3D masks that use filtered 3D Gaussian noise to produce smooth, mid-frequency spatiotemporal occlusions across frames. For audio, it introduces Regularized Blue Noise (R-BN), an optimization-based mask that ensures local separation of visible patches for uniform time, frequency coverage. The goal is to create structured masking patterns that align with the intrinsic structure of each modality without adding training cost or supervision.

Empirically, replacing standard random masking with these structured masks yields consistent improvements across a range of video, audio, and multimodal tasks. For video, Green3D enhances models such as VideoMAE and SIGMA by small but consistent margins on benchmarks like Kinetics-400 and Something-Something V2, and larger gains are seen on unsupervised video segmentation and generalization tasks. For audio, R-BN provides measurable boosts across AudioMAE, MaskSpec, and other baselines. In multimodal setups, such as with CAV-MAE, modality-specific masks improve both unimodal and joint performance. Overall, the method is simple, plug-and-play, and incurs no additional computational cost. However, while practical, the conceptual leap beyond prior colored-noise approaches is moderate, and the performance gains are incremental rather than transformative.

**Strengths:**

- Practical, modality-aware design: The proposed structured masks can be precomputed and integrated into existing frameworks without additional computation or parameters.

- Consistent cross-domain improvements: The approach improves performance across video, audio, and multimodal benchmarks, demonstrating its generality.

- Strong gains in structure-sensitive settings: The largest benefits appear on tasks requiring spatiotemporal continuity or fine-grained structural reasoning.

- Clear visualizations and ablations: The paper includes intuitive visual examples and quantitative analyses explaining why certain noise spectra (green vs. blue) are better suited for specific modalities.

- Good documentation and reproducibility: Implementation details, pseudo-code, and experimental settings are clearly described, making the approach easy to reproduce.

**Weaknesses:**

- Incremental novelty: The main conceptual step, extending colored-noise masking from 2D to 3D and to audio, is a straightforward generalization of previous ideas, with limited theoretical innovation.

- Small headline gains: On standard benchmarks, the improvements are modest and may not justify publication at a top-tier venue without stronger theoretical or empirical justification.

- Comparability issues: Some evaluations rely on custom dataset splits or self-reimplemented baselines, which limit direct comparison to prior work.

- Limited theoretical grounding: The paper motivates structured noise heuristically but lacks deeper analysis of why these patterns yield better learning or reconstruction properties.

- Incomplete robustness evaluation: There is little exploration of sensitivity to mask ratios, cross-distribution generalization, or behavior on longer videos and non-speech audio.

**Questions:**

NA

---

> ### Author Response · Authors · 2025-11-26
>
> We appreciate the insightful and constructive review, especially for recognizing our "practical, modality-aware design," the "consistent cross-domain improvements," and the "strong gains in structure-sensitive settings" enabled by our masking strategy. We address your remaining concerns point by point below and provide clarifications and additional results where helpful.
>
> -------------------------
>
> **W1 — Incremental novelty: extending colored-noise masking from 2D to 3D/audio is seen as a straightforward generalization with limited theoretical innovation.**
>
> We agree with the reviewer that our work is inspired by colored-noise masking (Hinojosa et al., 2024), yet our study demonstrates that extending ColorMAE to video and audio is not a straightforward generalization. Both video and audio impose structural, modality-specific constraints that make simply applying naive 2D color masks suboptimal, as reflected in our ablations.
>
> **Video.** Copying 2D ColorMAE masks across frames or sampling them independently creates temporal discontinuities, which result in suboptimal downstream performance.  This underscores the need for video modality-aware masking that produces temporally coherent occlusions aligned with natural video statistics, motivating the proposed Green3D masks. This is empirically validated in Table 7 of the main paper, also provided in the table below, which shows 2D green masks (ColorMAE) considerably underperform our proposed Green3D masking on both mini-SSv2 (52.9 vs 54.5) and mini-Kinetics (51.9 vs 52.7).
>
> | Masking Type | mini-SSv2 | mini-Kinetics |
> |-------|------|--------|
> | Green2D    | 52.9 | 51.9   |
> | Green3D    | 54.5 | 52.7   |
>
> **Audio.** Spectrograms are highly anisotropic and structured along harmonic bands. Table 6 shows that red/green masks remove entire time-frequency regions, thereby harming AudioMAE performance. The proposed Regularized Blue Noise (R-BN) introduces a new optimization objective (directional separation + patch-spacing constraints) to ensure uniform coverage of harmonic structure, a crucial audio property. This is empirically validated in Table 7 of the main paper, also provided in the table below, which shows blue masks (following ColorMAE) underperform our proposed R-BN masking on both AS-20k (36.5 vs 36.8) and ESC-50 (94.2 vs 94.6).
>
> | Masking Type | AS-20k | ESC-50 |
> |-------|------|--------|
> | Blue    | 36.5 | 94.2   |
> | R-BN    | 36.8 | 94.6   |
>
>
> **Beyond ColorMAE.** To further verify that our improvements do not simply arise from using *any* structured mask, we additionally evaluated common non-learned masking strategies used in prior work, including grid, block, and tube masking, within the same VideoMAE setup. Consistent with reports in SimMIM/MAE and related literature, these fixed spatial patterns underperform random Tube masking, and in our experiments, they also fall short of Green3D:
>
> | Masking Type | mini-SSv2 | mini-Kinetics |
> |--------------|-----------|---------------|
> | Grid         | 52.3      | 51.0          |
> | Block        | 52.5      | 51.1          |
> | Tube         | 52.8      | 51.6          |
> | Green3D      | 54.5      | 52.7          |
>
> These results reinforce the idea that trivial structured masks, such as grid and block patterns, do not improve masked modeling, which explains why random masking remains the standard in MAE-based methods. To our knowledge, our method is the first data-independent masking strategy to consistently outperform random masking across video, audio, and multimodal settings without added computation, reflecting a principled mask design for spatiotemporal and time-frequency domains rather than a simple 2D extension. We have emphasized this within the contributions section of our paper and added the grid/block masking ablation in Appendix A.4 of the revised manuscript.

---

> ### Author Response · Authors · 2025-11-26
>
> **W2 — Improvements on standard benchmarks are modest**
>
> Thank you for the opportunity for clarification. In masked modeling methods, improvements of 0.5-1% on large-scale recognition benchmarks are considered meaningful, especially without significant architectural changes or computational overhead (as reported in MAE, VideoMAE, SimMIM, and SIGMA). In this context, the +0.7–1.2% improvements on datasets like  Kinetics-400 and Something-Something V2 are already on the upper end of what masking-level changes achieve, especially given our zero computational overhead.
>
> We would also like to highlight that adding just our masking to the vanilla VideoMAE architecture (VideoMAE+Green3D) yields substantial gains on structure-sensitive tasks, e.g., +8.7% on DAVIS and 1.5% on YTVOS unsupervised segmentation (Table 2), outperforming methods like MME and SIGMA, which already make considerable design changes in the masked video modeling architecture. Similarly, an improvement of +1–7.0% on the SEVERE generalization benchmarks is worth highlighting (Table 3). This shows our modality-aware masking introduces meaningful inductive biases that random masking lacks and can act as a simple plug-and-play module for both existing and future masked modeling methods.
>
> ---------------
>
> **W3 — Comparability concerns due to custom splits and reimplemented baselines, limiting direct comparison to prior work.**
>
> We apologize for the confusion caused by the presentation of our results. We clarify that all headline results use the official implementations and training protocols of VideoMAE, SIGMA, AudioMAE, and CAV-MAE. Custom subsets appear only in small-scale ablations (e.g., Tables 6–7) to compare many masking variants uniformly; these do not affect any of our main results.
>
> **Video.** All results on Kinetics-400, Something-Something V2, DAVIS-2017, and SEVERE strictly follow the standard splits and codebases used in the respective papers used for comparison.
>
> **Audio.** For AudioMAE, we use the original authors' pretrained checkpoints and standard AS-2M, AS-20k, and ESC-50 splits. As discussed in Appendix A.13, the only deviation is during evaluation on AudioSet: YouTube periodically removes videos, and the exact test split used in the AudioMAE paper is no longer available. We therefore follow the widely adopted, publicly available newer AudioSet test set (Hugging Face). Importantly, both AudioMAE and our model are evaluated on the same publicly available test split, ensuring a fair comparison; differences from the numbers reported in the AudioMAE paper stem from their use of an older split that is no longer accessible. We have modified the manuscript in section 4.2 to highlight this.
>
>
> **Audio–Video.** For CAV-MAE, the original paper reports AudioSet results, but AudioSet cannot be fully downloaded due to IP and copyright restrictions. As described in Appendix A.14, we rely on the authors' public training pipeline for VGGSound and evaluate both the baseline and our method under identical settings. As expected, absolute numbers differ from those in the original paper, but we believe this setup provides a controlled and fair comparison. For future-proof comparisons, we will make all source code, model checkpoints, and dataset splits available on our project website. We have modified the manuscript in section 4.3 to highlight this.
>
> **Consistency.** Across all official and reproduced settings, including video, audio, and audio-video tasks, we observe consistent improvements over the random-mask baseline. This consistency across independent implementations and varying datasets supports that the gains arise from the proposed masking strategy itself rather than from preprocessing or custom splits.

---

> ### Author Response · Authors · 2025-11-26
>
> **W4 — Deeper Analysis of Mask Choice per Modality**
>
> We appreciate your feedback. In the revised Appendix, we added a deeper analysis to complement our heuristic intuition. First, we conducted a frequency-domain study (Fig. A.6) showing why our structured masks are appropriate for each modality. Specifically, Blue masks are high-frequency, which means they remove patches in a fine-grained and uniformly distributed manner. This prevents large contiguous occlusions that would otherwise destroy the harmonic structure of audio spectrograms, which are dominated by low-frequency energy. On the other hand, Green3D masks match the low-to-mid–frequency profile of natural video frames and are better for this modality because they retain more semantically informative spatio-temporal structures, aligning with findings in [A].
>
>
> | Masking        | 400  epochs           | 600  epochs           | 800   epochs          |
> |----------------|-----------------|-----------------|-----------------|
> | Random         | 70.283 ± 0.34   | 77.063 ± 0.167  | 76.073 ± 0.683  |
> | Green3D        | 105.029 ± 0.777 | 105.364 ± 0.437 | 107.656 ± 0.640 |
>
>
>
> Second, as established in [B], standard MAE often suffers from dimensional collapse, in which a lower effective rank indicates that features are confined to a low-dimensional subspace. Higher effective rank, conversely, indicates better learning dynamics and more diverse feature representations. Following [B], we perform an experiment to compute the effective rank in VideoMAE. This is calculated by normalizing the singular values of the feature matrix to a probability distribution and then taking the exponential of its Shannon entropy. Essentially, it measures how many significant dimensions the feature representation actually utilizes. We report the results in the table above using VideoMAE pretrained with Random masking and our proposed Green3D. We observe that the effective rank with Green3D is substantially higher than with Random masking, indicating it produces richer feature representations. These theoretical findings are consistent with the downstream task performance reported in our paper and appendix.
>
> [A] Haochen Wang, et al. "Hard patches mining for masked image modeling." *Proceedings of the IEEE/CVF Conference on Computer Vision and Pattern Recognition*. 2023.
>
> [B] Qi Zhang, et al. "How mask matters: Towards theoretical understandings of masked autoencoders." *Advances in Neural Information Processing Systems* 35 (2022): 27127-27139.
>
> --------------------
>
> **W5 — Limited robustness analysis, with insufficient evaluation of mask ratio sensitivity and cross-distribution/generalization behavior.**
>
> Thank you for this suggestion. We would like to highlight that several robustness analyses are already included in our paper; however, we agree that their presentation could have been clearer. We have clarified them in the revised version and summarized them here.
>
> **Mask ratio and hyperparameter sensitivity.** Appendix A.3 provides comprehensive mask-ratio ablations, showing that performance remains stable across 70–90% masking, with downstream accuracy varying by less than 0.5%. Appendix A.1 reports $\sigma_1$–$\sigma_2$ sweeps for Green3D, demonstrating that randomized mid-frequency ranges consistently achieve the best results, while extreme values degrade performance. Together, these findings indicate that our method is not highly sensitive to hyperparameter choices.
>
>
> **Cross-distribution generalization.** Our structured masks generalize well to out-of-distribution and long-horizon video settings. The SEVERE benchmark (Table 3) measures cross-distribution robustness and shows clear improvements over the random baseline on all categories. We have modified the manuscript to highlight this.
>
> **Longer videos and non-speech audio.** VideoMAE samples 16-frame clips from videos exceeding 200+ frames; our masks are applied per clip and are shown to improve both short- and long-horizon tasks (e.g., DAVIS, SEVERE). Audio experiments include AS-20k and VGGSound, both of which contain large fractions of non-speech audio. We have modified the manuscript to highlight this.
>
> ---------------------------
>
> Thank you again for the constructive feedback. We have added new empirical evidence, deeper analysis, and clearer justification addressing the concerns about novelty, comparability, theoretical grounding, and robustness.  We hope these additions further strengthen the contribution and practical relevance of our modality-aware masking design, and we would appreciate your consideration of an updated evaluation if you feel the concerns have been satisfactorily resolved.

---

### Author Response · Authors · 2025-12-03

We sincerely thank all reviewers for their time and constructive feedback. The reception has been encouraging, with consistent strengths noted across reviews, including the “practical, modality-aware design” (**NX9G**), “consistent cross-domain improvements” (**NX9G**), the “simple and effective” data-independent masks (**iQhj**), “strong compatibility in multimodal settings” (**iQhj**), the “impressive” results (**1dtC**), and the recognition of our “solid contribution towards better masking” (**KjQ5**). Reviewers also highlighted the clarity of presentation, visualizations, ablations, and reproducibility.

Below we summarize how each major concern was addressed:

**1. Novelty and modality-specific challenges**
We clarified why direct 2D ColorMAE extensions underperform and provided explicit comparisons along with a detailed explanation of the structural, modality-specific constraints motivating Green3D and R-BN resolving Reviewer **1dtC**'s main concern.

**2. Signal-processing motivation**
We added a frequency-domain and effective-rank analysis (Fig. A.6, Sec. A.15) showing why mid-frequency masks suit video and high-frequency, uniform masks suit audio to address intuition/theory questions raised by **NX9G** and **iQhj**.

**3. Additional baselines and ablations**
We surfaced ablations already present in the appendix (mask-ratio sweeps; Green3D σ₁–σ₂ sensitivity) and added the small targeted analyses specifically requested:
- Grid, block, and tube masking baselines (A.4)
- R-BN hyperparameter sensitivity (Δ and directional weights) (A.2)
- Multi-seed variance analysis (A.7)

**4. Clarifications of setups**
We clarified the AudioSet split issue, the CAV-MAE training protocol, the UCF-RC metric directionality, and added the missing SPC-2 citation.

We hope this summary and the detailed responses below address all reviewers' concerns and help clarify the contribution and scope of our work. We appreciate the reviewers’ constructive input and the opportunity to further strengthen the paper.

---

### Meta-Review · Area_Chair_qofU · 2025-12-29

**Summary:**

The strengths of the paper are consistently recognized across the reviews. Reviewer NX9G highlights a “practical, modality-aware design” where “the proposed structured masks can be precomputed and integrated into existing frameworks without additional computation or parameters,” and notes “consistent cross-domain improvements” with “strong gains in structure-sensitive settings,” along with “clear visualizations and ablations” and “good documentation and reproducibility.” Reviewer KjQ5 writes that “overall, this paper has a solid contribution towards better masking for Video and audio,” that “the authors have presented results with strong baselines showing improvements on standard benchmarks,” and that “the results are particularly impressive for SEVERE benchmark, showing the generalization,” while emphasizing that “the data independent approach of Green 3D makes it much more efficient than alternative approaches that required motion prior.” Reviewer 1dtC states that “the paper organization is easy to follow,” that “the mask modeling methods are understandable,” and that “the experiment results are impressive.” Reviewer iQhj notes that “the method is both simple and effective: it enhances self-supervised learning performance across multiple modalities using predefined noise structures, without relying on data or external supervision,” and that “extensive experiments have been conducted” and that “the paper is well-structured and clearly articulated.”

The 4 reviewers recommend 6, 8, 4, 4 ratings;the AC recommends rejection because, although this is a careful, well-executed piece of work and a clean design, the perceived conceptual and empirical advance over existing works is still modest relative to ICLR’s bar. The Ac notes that all the reviewers, including the positive ones point out limited novelty / incremental nature. The AC is especially concerned about the current works' utilization for and impact on future work. The empirical results show consistent gains, but on a rather marginal level, which can typically also be achieved by using more advanced data augmentation schemes. The reviewers are also not providing clear reasons on why this work would make an impact in the community. Reviewer NX9G, with rating marginally above the acceptance threshold but would not mind if paper is rejected, emphasizes “incremental novelty” and “small headline gains” on standard benchmarks, and questions whether this is enough without stronger theoretical or empirical justification. Reviewer KjQ5, with accept rating sees a “solid contribution towards better masking” and is mainly concerned with incremental nature and specific ablations; this is the most positive review (their main reasons to accept the paper are not very specific). Reviewer 1dtC, with marginally below the acceptance threshold, but would not mind if paper is accepted, explicitly flags “limited novelty” as the primary issue. Reviewer iQhj, also with same rating focuses on missing theoretical justification and robustness studies, which the authors have attempted to address post‑hoc. The updated appendix material (frequency‑domain analysis, effective rank, mask-ratio sweeps, hyperparameter and multi‑seed variance studies, grid/block baselines, clearer metric interpretation) substantially improves the technical completeness and would likely lead to somewhat more favorable views from the more critical reviewers. However, even with these additions, the core contribution remains a carefully tuned, modality‑aware structured masking scheme built on colored noise, with gains that, while consistent and sometimes sizable on specific structure‑sensitive benchmarks, are still quite modest on main video and audio benchmarks (and in the AC's view similar gains can also be achieved with more advanced data augmentation techniques or hyperparameter tuning). In this setting, one strongly positive review, but without clear indication why the work is worth accepting, and three marginal reviews (two below and one slightly above the threshold, all saying they “would not mind if paper is accepted/rejected”) do not provide a strong enough consensus to support acceptance.
On balance, the AC sees no basis to accept the paper given reviewer suggestions, and their own view after reading the paper as well as the author rebuttal/discussions. The paper presents a simple, data‑independent structured masking strategy that consistently improves the baselines, but its novelty is seen by several reviewers as incremental over ColorMAE, and even with added frequency‑domain analysis and robustness ablations, the work does not yet offer the level of conceptual insight or transformative empirical impact expected at ICLR. The AC highly recommends the authors to address the concerns of the reviewers and take into account their suggestions of improvement when preparing a revised version. The AC further notes that some accuracy gains in the paper's tables are incorrectly calculated (the blue +% numbers in the tables).

**Reviewer Concerns:**

The weaknesses focus mainly on novelty, depth of analysis, and completeness of the empirical study. Reviewer NX9G characterizes the work as having “incremental novelty,” describing the main step as “extending colored-noise masking from 2D to 3D and to audio” which is seen as “a straightforward generalization of previous ideas, with limited theoretical innovation,” and also points to “small headline gains” and “limited theoretical grounding” and “incomplete robustness evaluation.” Reviewer KjQ5 remarks that “the paper contribution comes across as a bit incremental” and asks for clarification of specific comparisons (MGM/MGMAE vs the proposed approach) and for “some ablation study done varying target masking ratio using these new masking approaches.” Reviewer 1dtC lists “limited novelty,” writing that “this paper seems to apply the ColorMAE approach to audio and video tasks” and that “the authors should emphasize what specific challenges arise when using a color-based MAE for audio and video, and how these challenges are addressed,” and also notes that “some details are unclear,” including why a particular masking strategy in Figure 3 pearforms better. Reviewer iQhj points to “lack of theoretical and intuitive justification,” “insufficient ablation studies and analysis” including hyperparameter sensitivity, missing comparisons to other structured masks like grid or block, concerns about performance variance, and mentions “unexplained experimental results” such as the way UCF‑RC is reported.
What seems to be missing in the submission, even after the current revisions, and besides stronger absolute empirical evidence, is a stronger case for conceptual novelty beyond being a well-engineered extension of ColorMAE and a clearer theoretical framing of why the specific color-noise choices are principled for each modality. Reviewer NX9G explicitly asks for “deeper analysis of why these patterns yield better learning or reconstruction properties” and for more exploration of “sensitivity to mask ratios, cross-distribution generalization, or behavior on longer videos and non-speech audio.” Reviewer 1dtC requests explicit “ablation study comparing ColorMAE with the proposed method” and a cleaarer explanation of the modality‑specific challenges. Reviewer iQhj asks for “a more in-depth explanation from a signal processing perspective” and presses for systematic hyperparameter analysis, comparisons with grid/block masking, multi‑seed variance, and discussion of cases where performance drops. While many of these are addressed in the appendix according to the author comments, some of the key conceptual points (perceived incrementality, limited theoretical innovation) remain more a matter of framing than of new core ideas, and the main paper still leans heavily on empirical gains that, while consistent, are modest in more popular evaluations/benchmarks

**Reviewer Scores:**

The author comments are thorough and address many of the specific technical concerns raised by the reviewers, but they do not fully remove the perception of limited novelty. In response to Reviewer NX9G, the authors argue that extending ColorMAE to video and audio is “not a straightforward generalization,” present tables where Green2D and Blue masks underperform Green3D and R‑BN, and add grid/block/tube baselines showing that naive structured masks underperform Green3D. They also highlight that +0.5–1% gains are meaningful in MAE‑style settings (this is questionable; for 1600 epochs ImageNet training schedules this may be true but not for shorter duration training and evaluation on more noisy, and espeically less popular, video benchmarks); Authors are further poainting out strong improvements on SEVERE and unsupervised segmentation, and clarify dataset split issues and evaluation protocols. They further add frequency‑domain and effective-rank analyses to reinforce the signal-processing motivation and robustness studies (mask ratios, hyperparameters, SEVERE generalization, multi‑seed variance) to respond to Reviewer iQhj’s and Reviewer KjQ5’s concerns. For Reviewer 1dtC, they explicitly compare ColorMAE (Green2D/Blue) to Green3D/R‑BN and give more qualitative analysis of why R‑BN improves over random masking, clarifying the difference from simply “applying ColorMAE” to other modfalities. However, the reviewers did not have the opportunity to update their ratings after these comments. Based on the nature of the concerns and the responses, it is plausible that Reviewer 1dtC, who wrote “I will increase my rating if weakness 1 is solved,” might modestly increase their score, since an explicit ColorMAE vs Green3D/R‑BN comparison and more explanation has been provided in rebuttal. Reviewer iQhj’s requests for theory, hyperparameter ablations, grid/block baselines, multi‑seed variance, and the missing SPC‑2 citation appear largely addressed in the revised appendix, so their concerns about robustness and compleateness may also be partly alleviated. Reviewer NX9G’s reservations about incremental novelty and modest headline gains are more about the overall conceptual weight; the new analyses improve grounding but likely do not transform the perceived contribution. Overall, the author respones strengthen the paper and address many detaled questions, but they do not clearly change the fundamental balance between solid engineering and limited innovation.

---

### Decision · Program_Chairs · 2026-01-26

Reject